# Photochemically-Induced Release of Lysosomal Sequestered Sunitinib: Obstacles for Therapeutic Efficacy

**DOI:** 10.3390/cancers12020417

**Published:** 2020-02-11

**Authors:** Judith Jing Wen Wong, Maria Brandal Berstad, Ane Sofie Viset Fremstedal, Kristian Berg, Sebastian Patzke, Vigdis Sørensen, Qian Peng, Pål Kristian Selbo, Anette Weyergang

**Affiliations:** 1Department of Radiation Biology, Institute for Cancer Research, The Norwegian Radium Hospital, Oslo University Hospital, 0379 Oslo, Norway; judwon@rr-research.no (J.J.W.W.); mberstad@rr-research.no (M.B.B.); anfrem@rr-research.no (A.S.V.F); kristian.berg@rr-research.no (K.B.); sebastian.patzke@rr-research.no (S.B.); selbo@rr-research.no (P.K.S.); 2Section for Pharmaceutics and Social Pharmacy, Department of Pharmacy, University of Oslo, 0371 Oslo, Norway; 3Department of Core Facilities and Department of Molecular Cell Biology, Institute for Cancer Research, The Norwegian Radium Hospital, Oslo University Hospital, 0379 Oslo, Norway; vigdis.sorensen@rr-research.no; 4Department of Pathology, The Norwegian Radium Hospital, Oslo University Hospital, 0379 Oslo, Norway; qian.peng@rr-research.no

**Keywords:** sunitinib, photochemical internalization, photodynamic therapy, lysosomal sequestration, cytosolic release, photochemical release, sunitinib resistance

## Abstract

Lysosomal accumulation of sunitinib has been suggested as an underlying mechanism of resistance. Here, we investigated if photochemical internalization (PCI), a technology for cytosolic release of drugs entrapped in endosomes and lysosomes, would activate lysosomal sequestered sunitinib. By super-resolution fluorescence microscopy, sunitinib was found to accumulate in the membrane of endo/lysosomal compartments together with the photosensitizer disulfonated tetraphenylchlorin (TPCS_2a_). Furthermore, the treatment effect was potentiated by PCI in the human HT-29 and the mouse CT26.WT colon cancer cell lines. The cytotoxic outcome of sunitinib-PCI was, however, highly dependent on the treatment protocol. Thus, neoadjuvant PCI inhibited lysosomal accumulation of sunitinib. PCI also inhibited lysosomal sequestering of sunitinib in HT29/SR cells with acquired sunitinib resistance, but did not reverse the resistance. The mechanism of acquired sunitinib resistance in HT29/SR cells was therefore not related to lysosomal sequestering. Sunitinib-PCI was further evaluated on HT-29 xenografts in athymic mice, but was found to induce only a minor effect on tumor growth delay. In immunocompetent mice sunitinib-PCI enhanced areas of treatment-induced necrosis compared to the monotherapy groups. However, the tumor growth was not delayed, and decreased infiltration of CD3-positive T cells was indicated as a possible mechanism behind the failed overall response.

## 1. Introduction

Small-molecule inhibitors such as tyrosine kinase inhibitors (TKIs) target intracellular signal pathways and are designed to cross cellular membranes passively due to their size and lipophilicity. Inside the cell they may exert their action by blocking the ATP-binding pocket of their target protein, thereby inhibiting protein signal transduction. Drug distribution within the cell is therefore an important consideration for drug efficacy. Sunitinib is an orally administered, multi-targeted TKI approved by the United States Food and Drug Administration and the European Medicines Agency for the treatment of imatinib-resistant gastrointestinal stromal tumor, advanced renal cell carcinoma and advanced pancreatic neuroendocrine tumors [1]. Sunitinib exerts its anti-cancer effect directly both on the tumor vasculature and tumor cells [2]. In addition, sunitinib has been shown to inhibit tumor growth by inducing anti-tumor immunity, through reduction of both myeloid-derived suppressor cells (MDSCs) and T regulatory cells (Tregs) [3,4]. The main targets of sunitinib are vascular endtothelial growth factor receptors (VEGFRs), platelet-derived growth factor receptors (PDGFRs), fms-related tyrosine kinase 3 (FLT-3), the stem cell factor receptor (KIT) and the colony stimulating factor 1 (CSF-1), which all are inhibited at nanomolar concentrations [5]. Sunitinib is a weak base (pKa = 8.95, Appendix A) and is therefore subjected to increased protonation at decreasing pH values in the physiological pH span. Consequently, protonated sunitinib is sequestered in acidic lysosomes, which impact its intracellular distribution and may reduce its target interaction and, hence, drug efficacy [6,7]. Lysosomal sequestration of sunitinib has previously been shown in acquired sunitinib-resistant HT-29 colon cancer cells [7]. The impact of sequestration on sunitinib resistance has, however, not been studied. Lysosomal sequestration is not only limited to sunitinib, but is also observed for other anti-cancer weak-basic drugs such as other TKIs (gefitinib, lapatinib [8]), anthracyclines (doxorubicin [9], daunorubicin [10]) and preclinical imidazocridinones [6,11,12].

Photochemical internalization (PCI) represents a clinically relevant treatment modality for release of drugs that are entrapped in endosomes and lysosomes [13,14,15]. This method is based on amphiphilic photosensitizers (PSs), such as meso-tetraphenylchlorin disulfonate (TPCS_2a_) (Appendix A), which localize in the membrane of endocytic vesicles [14]. Upon illumination, reactive oxygen species (ROS) generated during photochemical reactions destabilize the vesicle membrane and induce cytosolic release of the content entrapped in these vesicles [16,17]. Drugs which accumulate in endo/lysosomal compartments can in this way be released to interact with their intracellular target. PCI may be applied with two different protocols; in the “light after” protocol light exposure is applied after administration of the drug to be released, while in the “light first” protocol the photochemical reaction is generated before administration of the drug [18]. PCI was first documented as a delivery technology of hydrophilic macromolecules which do not readily penetrate the plasma membrane. Recent evidence has, however, indicated the technology as efficient also for small molecules entrapped in endocytic vesicles, such as gemcitabine and doxorubicin [13,19,20,21]. Here, we evaluated PCI as a strategy to release lysosomal sequestered sunitinib, thereby enhancing the cytotoxicity of sunitinib in cancer cells.

## 2. Results

### 2.1. Lysosomal Localization of Sunitinib and TPCS_2a_

PCI is dependent on endo/lysosomal localization of both the PS and the molecule subjected to cytosolic release. Fluorescence microscopy of sunitinib and TPCS_2a_ was therefore employed to evaluate the intracellular localization of both compounds. Granular fluorescence of sunitinib was detected after 24 h incubation in HT-29 cells (Figure 1a). The fluorescence was to a large degree overlapping with LysoTracker Red in agreement with a previous report by Nowak-Sliwinska et al. [22], indicating accumulation of sunitinib in endo/lysosomal vesicles. TPCS_2a_ enters the cell by means of adsorptive endocytosis and accumulates in the membrane of endo/lysosomal vesicles [13,16]. An 18 h incubation of TPCS_2a_ resulted in granular fluorescence, partly overlapping with LysoTracker, indicating endo/lysosomal localization in HT-29 cells (Appendix A). Fluorescence was, however, also detected on the plasma membrane indicating non-endocytosed PS. Plasma membrane associated TPCS_2a_ was largely removed by a 4 h incubation in TPCS_2a_-free medium, resulting in an overall granular pattern of fluorescence. Thus, TPCS_2a_ accumulates mainly in endo/lysosomes in HT-29 cells when used as in a standard PCI protocol (Figure 1b, left panel) in agreement with previous reports [14,16]. Photochemical destabilization of the endo/lysosomal membrane and subsequent cytosolic release of the entrapped drug of interest is the basic mechanism behind PCI. Light exposure was here shown to relocalize TPCS_2a_ to the cytosol in HT-29 cells, which corresponded with a reduction of the LysoTracker signal (Figure 1b, right panel).

### 2.2. No Enhanced Toxicity by PCI “Light After” of Sunitinib

The fluorescence images in Figure 1a,b reveal endo/lysosomal localization of both TPCS_2a_ and sunitinib, and it was therefore expected that light activation of the photosensitizer would result in cytosolic release of sunitinib. This PCI protocol was in accordance with the PCI “light after” procedure where light exposure is applied after administration of the drug to be released. No enhanced cytotoxicity was, however, indicated following PCI of sunitinib exposing the cells to blue light from LumiSource™ (Figure 1c). The observed combined effect was found to be slightly higher than the theoretical additive effect, and the synergy/antagonism parameter difference in log (DL) indicated a negative value −0.089 ± 0.075 although not significantly different from additively (*p* = 0.367). The absorption maximum for sunitinib has previously been reported at 429 nm, which is near the maximum emission wave length of blue light source (λ_max_ = 437 nm) [22]. The PS TPCS_2a_ is also activated at its secondary maxima λ = 652 nm, allowing the circumvention of a putative blue-light induced inactivation of sunitinib. However, no increase in cytotoxicity was observed by PCI of sunitinib with the red light source (Figure 1d) yielding a slightly negative DL value −0.023 ± 0.09 (*p* = 0.834, not significant). PCI of the protein toxin gelonin (rGel) was included as a positive control for the PCI procedure, which resulted in synergistic cytotoxicity between rGel and the photochemical treatment, supported with a positive DL 0.262 ± 0.0048 (*p* < 0.001) (Figure 1e). Hence, PCI “light after” does not enhance the efficacy of sunitinib.

### 2.3. Sunitinib Is a Target for ROS-Mediated Photodamage

We investigated if the lack of enhanced cytotoxicity of sunitinib-PCI (“light after”) could be explained by ROS mediated photodamage of sunitinib. Singlet oxygen (^1^O_2_) is considered as the most important ROS formed during photochemical treatment as applied in this work [23,24]. The short half-life (<0.04 µs) and diffusion distance (10–20 nm) of singlet oxygen in cellular membranes [25] implicate that TPCS_2a_ should be in close intracellular vicinity of sunitinib in order to induce photochemical damage of the TKI. Super-resolution microscopy was therefore performed in order to evaluate the subcellular/suborganellar localization of TPCS_2a_ and sunitinib in detail. TPCS_2a_ and sunitinib partially co-localized in ring-like structures in single optical sections, indicating both compounds to be associated with vesicular membranes (Figure 1f). These results are in support for ROS-mediated photochemical damage of sunitinib.

Photodamage of sunitinib in the presence of TPCS_2a_ was further evaluated by absorption and fluorescence spectroscopy in solutions at pH 7 containing 1% fetal bovine serum (FBS) to solubilize these compounds. The emission spectra of both sunitinib and TPCS_2a_ prepared without light exposure were attenuated when they were combined (Figure 1g). However, the sunitinib fluorescence was reduced by approximately 50% while that of TPCS_2a_ was only reduced by ~20% (Appendix A). The fluorescence spectrum of sunitinib overlaps well the 4 Q-band absorption spectrum of TPCS_2a_ [16]. Thus, these results may be due to Förster resonance energy transfer (FRET) between sunitinib and TPCS_2a_ i.e., emission from sunitinib is absorbed by TPCS_2a_ and added to the directly excited TPCS_2a_. FRET may occur if the distance between the donor (sunitinib) and acceptor (TPCS_2a_) is short enough, typically 1–10 nm and is in line with the close proximity of the drugs in endo/lysosomal membranes [26]. The light exposure of both sunitinib and TPCS_2a_ separately lead to a smaller attenuation of sunitinib fluorescence (28%) than of TPCS_2a_ fluorescence (57%) (Figure 1g, table). When sunitinib and TPCS_2a_ was combined and exposed to light the TPCS_2a_ fluorescence was reduced to the same extent as in the absence of sunitinib, while the reduction in sunitinib fluorescence was much stronger in the presence of TPCS_2a_. These results indicate that the photooxidation of TPCS_2a_ is independent of the presence of sunitinib, while photoactivation of TPCS_2a_ contribute to the photooxidation of sunitinib. Similar results were observed with the absorption spectra where the broad absorption peak, 330–495 nm of sunitinib disappeared after light exposure of TPCS_2a_ (Appendix A).

Photochemical targeting of the endosomal membrane has previously been shown to release H^+^ from the endosomal lumen and thereby increase the endosomal pH prior to escape of endocytosed molecules [27] and pH 7 was therefore selected as the experimental condition. Nevertheless, the absorption spectra were also collected at pH 5 showing similar results as for pH 7 (Appendix A). These results therefore imply that sunitinib is photodamaged after light exposure of TPCS2a, most likely due to single oxygen-induced photooxidation.

### 2.4. Synergistic Cytotoxicity by “Light First” Sunitinib-PCI

PCI has usually been performed as in Figure 1c,d with the photochemical reaction initiated after administration of the drug of interest in combination with PS (“light after” procedure). However, PCI of non-targeting therapeutics has also been shown just as effective when the drug is administrated after the photochemical treatment (“light first” procedure) [17,18]. The suggested mechanism behind the “light first” procedure is fusion between photochemically damaged vesicles and intact vesicles containing the drug of interest [18]. Circumvention of the photochemical destruction of sunitinib during sunitinib-PCI was therefore attempted by applying PCI with the “light first” procedure. Indeed, treatment of HT-29 cells with sunitinib immediately after TPCS_2a_ incubation and light exposure was found to induce a synergistic reduction of viability as measured by the MTT assay (Figure 2a). The survival after sunitinib-PCI was found to be significantly lower than the theoretical additive effect at 8 µM sunitinib (*p* = 0.015) (Figure 2b) supported by a synergy/antagonism parameter DL of 0.84 ± 0.49 (*p* = 0.0035), indicating synergism. The effect of sunitinib-PCI was further verified by clonogenic assay indicating reduction of clonal cell survival by the PCI treatment (Figure 2c). The “light first” procedure was also applied to CT26.WT cells confirming enhanced reduction in cell viability by PCI of sunitinib (Figure 2d). At 2 µM sunitinib in CT26.WT, the DL value was 0.32 ± 0.046 (*p* = 0.019). Thus, PCI with the “light first” procedure enhances sunitinib cytotoxicity in both HT-29 and CT26.WT colon cancer cell lines in a synergistic manner.

Fluorescence imaging was performed to confirm cytosolic release of sunitinib (Figure 2e). Cytosolic release of sunitinib should result in a change in fluorescence from granules (endocytic vesicles) to diffuse fluorescence from the cytoplasm. The PCI “light after” protocol, with the photochemical treatment applied after sunitinib incubation, showed little difference in the fluorescence pattern of sunitinib upon light exposure, indicating poor cytosolic release with this procedure.

This was in contrast to the PCI “light first” protocol where a pronounced cytosolic fluorescence from sunitinib was observed upon light exposure. Thus, PCI with the “light first” procedure was indicated to release sunitinib from endosomal compartments, while PCI with the “light after” procedure was not.

### 2.5. Acquired Resistance and Endo/Lysosomal Accumulation Following Prolonged Sunitinib Exposure

Lysosomal accumulation of sunitinib has been reported as a mechanism of resistance [7,28,29,30,31,32] and we tested if PCI could counteract this mechanism by releasing endosomal sunitinib sequestered during long-term exposure. Sunitinib resistant HT-29 cells, HT-29/SR, was generated by continuous exposure to 2 µM. The low drug dose of 2 µM sunitinib reduced the viability in parental cells by approximately 20% (Figure 3a). The sunitinib concentration required to reduce the cell viability by 40% was 1.8 ± 0.08 (*p* < 0.001) fold higher in HT-29/SR cells compared to the parental cells. The sunitinib response in HT-29/SR cells measured by MTT assay was found similar within the time frame of these experiments indicating a stabilized sunitinib response in HT-29/SR cells (Figure 3a). The decreased sensitivity to sunitinib in HT-29/SR cells was demonstrated even more substantial by clonal cell survival as compared to overall cell viability (MTT) (Figure 3b). Overall, sunitinib sensitivity in HT-29 cells is higher than previously reported [7]. This is probably due to shorter incubation times (72 h versus 96 h).

The proliferation rate of HT-29/SR and HT-29 in the presence of sunitinib was also investigated (Figure 3c). A 2 µM sunitinib incubation for 120 h resulted in 42% confluence in the HT-29/SR cells while only 31% confluence was observed in the parental HT-29 cell line. The same tendency was shown for all sunitinib concentrations tested. Thus, the proliferation rate of HT-29/SR cells in the presence of 1–10 µM sunitinib was increased compared to the parental cells, further documenting the successful generation of sunitinib-resistant cells.

High degree of co-localization of sunitinib with LysoTracker Red confirmed lysosomal sequestering of sunitinib in HT-29/SR cells as verified by fluorescence microscopy (Figure 3d). The fluorescence images indicated higher fluorescence signals from sunitinib in HT-29/SR cells compared to the parental HT-29 cells (Figure 1a). Accumulation of sunitinib in HT-29 cells was further documented by flow cytometry, indicating a~1.8 fold higher accumulation in HT-29/SR cells that have been continuously incubated with sunitinib compared to parental cells subjected to 72 h sunitinib incubation (*p* = 0.03, one-tailed *p* value) (Figure 3e). Accumulated sunitinib in HT-29/SR cells was released when removing sunitinib from the media for 24 h (Appendix A). In agreement with this, there was no difference in sunitinib accumulation between HT-29 and HT-29/SR cells when a 24 h wash in drug-free medium was added before the 72 h sunitinib incubation (Appendix A).

### 2.6. Photochemical Release of Sequestered Sunitinib Does Not Abolish Sunitinib Resistance in HT-29/SR

As sunitinib was found subjected to photochemical damage by TPCS_2a_ and light, it was expected that sunitinib-PCI with “light after” would not induce synergistic effects on cell viability. Indeed, no difference in light-induced toxicity was found between TPCS_2a_-treated HT-29/SR and HT-29 cells (Figure 3f), and PCI with “light after” procedure yielded no statistically significant difference between sunitinib alone and PCI “light after” at 8 µM sunitinib (Figure 3g).

The antagonism/synergy DL parameter −0.161 ± 0.154 (*p* = 0.49, not significant) indicated additivity, but towards antagonism similar to the parental HT-29 cells (Figure 1c,d). PCI with the “light first” strategy induced reduction in HT-29/SR cell viability (Figure 3h, *p* = 0.044 at 8 µM sunitinib compared to PCI). The effect was further evaluated using the DL parameter and was at 8 µM sunitinib 0.049 ± 0.094 (*p* = 0.66, not significant) indicating that the effect was additive with this strategy. Thus, the sunitinib resistance was not abolished by the sunitinib concentration applied to reduce viability with neither “light first” nor “light after” PCI. Notably, PCI of rGel with “light after” strategy was found similarly efficient in both HT-29 (Figure 1e) and HT-29/SR implying rGel-PCI to circumvent sunitinib resistance (Figure 3i).

### 2.7. Modest HT-29 Tumor Growth Delay After Sunitinib-PCI in Athymic Mice

Based on the in vitro results, PCI with “light first” protocol, was selected for the in vivo evaluation of sunitinib-PCI in HT-29 xenografts in athymic mice. rGel-PCI has previously been reported as least as efficient with the “light first” procedure as with the “light after” procedure in vivo [17]. The “light first” effect is previously indicated up to ~8 h post-photochemical treatment [18] while sunitinib reaches a maximum plasma concentration 3–6 h post-oral administration [33]. In the first in vivo treatment protocol (Sun1-PCI) sunitinib was therefore administrated 3 h prior to and 30 min after light exposure. Fluorescence microscopy of frozen tumor sections revealed the presence of both TPCS_2a_ and sunitinib in the tumor at the point of light exposure (Figure 4a). The granular TPCS_2a_ fluorescence indicated localization in endosomes and lysosomes. The sunitinib fluorescence was, however, mostly weak and too diffuse to make any conclusions on intracellular localizations. However, some overlap in granular sunitinib and TPCS_2a_ fluorescence was detected in the frozen sections indicating in vivo co-localization. Light exposure of the tumors at 15 J/cm^2^ induced a reduction of both TPCS_2a_ and sunitinib fluorescence (Figure 4b), most likely reflecting photochemical damage of sunitinib as observed in vitro and photobleaching of TPCS_2a_ (Figure 1g) [16].

Sun1-PCI did not increase the overall treatment effect compared to PS + light only or Sun1 only, as shown in the Kaplan-Meier plot and in the estimated mean time to reach endpoint (tumor size < 900 mm^3^) (Figure 4c,e). In an effort to enhance the anti-tumor efficacy, we extended the duration of sunitinib treatment after light exposure by additional administration on day 1, 2, 3 and 4 post-light exposure (Sun2 protocol). The estimated time to reach endpoint was significantly increased compared to untreated controls in all groups receiving sunitinib, and the longest estimated time to reach endpoint was found for the Sun2-PCI-treated animals (25 d compared to 12 d for untreated animals) (Figure 4d,e). The growth curves of each individual animal indicated tumor growth delay the first 10 days after Sun1-PCI and Sun2-PCI (Appendix A). The average tumor size at day 6 and 10 after treatment was therefore assessed in all treatment groups (Figure 4f). The one-way ANOVA test revealed significant differences between the treatment groups at both time points (*p* < 0.001 at day 6, and *p* = 0.013 at day 10). Further pair-wise multiple comparison (Holm-Sidak method) indicated significant PCI-Sun1-, Sun2- and PCI-Sun2-induced tumor growth delay at 6 days compared to untreated controls (Figure 4g). The tumor volume of PCI-Sun1-treated animals was also significantly smaller than in Sun1-treated animals at day 6. However, no significant difference was found in tumor size between Sun1-PCI- and Sun2-PCI-treated animals at day 6 and 10. The only significant difference at day 10 was found between Sun1-PCI and untreated controls (Figure 4g). Thus, even though sunitinib-PCI showed some initial anti-tumor efficacy compared to non-treated controls, the overall treatment effect was less than expected. Since PCI in vitro could not circumvent sunitinib resistance, the HT-29/SR model was not continued in vivo.

### 2.8. Poor CT26.WT Tumor Growth Delay After Sunitinib-PCI in Immunocompetent Mice

We have previously shown that T-cell activation is important to achieve curative effects after PCI [34,35]. In addition, sunitinib is shown to stimulate anticancer immune response [3,4]. Hence, sunitinib-PCI was explored in CT26.WT tumor-bearing thymic mice. The overall tumor responses of sunitinib-PCI was, however, smaller in the CT26.WT model compared to the HT-29 model (Figure 5a,b). The normalized growth curves indicated a response at early time points (Appendix A). Although not significant, the estimated time to reach endpoint was found increased in all the treatment groups compared to untreated controls (Figure 5c). Early treatment responses were evaluated by comparing average tumor sizes in the different treatment groups at day 4 and 7 post-light exposure (Figure 5d,e). Significant difference among the treatment groups was found at day 4 (*p* = 0.015, one-way ANOVA), associated with a significant ~50% reduction in tumor volume in the Sun1-PCI group compared to untreated controls (Holm-Sidak method) (Figure 5e). No significant difference was found between any of the other treatment groups at day 4 (Figure 5e). Furthermore, no significant difference among the treatment groups was found at 7 d post-light exposure in this model (*p* = 0.216, one-way ANOVA) (Figure 5e).

### 2.9. Tumor Tissue Response

The poor response of sunitinib-PCI in the two tumor models was surprising in the light of the promising in vitro data on the “light first” procedure, especially in the CT26.WT model in immunocompetent mice where a strong treatment response was expected. The CT26.WT tumors at endpoint (900 mm^3^) revealed irregular shaped tumors in the PCI-treated animals as compared to the spherical shaped controls. Tumors following the Sun2-PCI protocol and corresponding controls were therefore harvested at endpoint and further evaluated for necrotic-, vascular- and immune-mediated treatment responses by hematoxylin and eosin (H&E) and immunohistochemistry (IHC) staining. Treatment-induced necrosis was detected in PS + light and Sun2-PCI-treated tumors compared to untreated and Sun2-treated tumors where no necrosis was observed (Figure 6a). The necrotic area in the Sun2-PCI-treated tumors were larger compared to in the PS + light group indicating more pronounced cancer parenchymal cell death in agreement with the in vitro results. The immunologic anti-tumor response of light activated TPCS_2a_ is dependent on T-cells infiltrating the treated area [34,36]. An increase in tumor-associated-CD3-positive T-cells was observed in both PS + light and Sun2-treated tumors compared to untreated controls (Figure 6b). Quantification of CD3-stained cells revealed a significant 5- and 3-fold increase in PS + light- and Sun2-treated sections respectively compared to untreated tumors (Figure 6c). No increase was, however, detected in Sun2-PCI-treated tumors (Figure 6c), indicating an antagonistic effect on tumor infiltrating T-cells in this treatment group.

A strong Ki-67 staining, indicating proliferating cells, was found in all viable areas in agreement with the H&E stains (Figure 6d), further indicating a stronger treatment response in Sun2-PCI-treated tumors compared to all the other treatment groups. TPCS_2a_ and light has previously been shown to target the tumor vasculature, and sunitinib efficacy has been strongly associated with a vascular response [16,37,38]. IHC with an αCD31 antibody was therefore applied to evaluate vascular responses of the combined treatment. A vascular treatment response was found in both photochemical-treated (PS + light), Sun2-treated and Sun2-PCI-treated tumor tissues, as visualized by rounded fragmented vessels compared to in the untreated controls. No additional effect was, however, found in the Sun2-PCI-treated tumors compared to the PS + light and Sun2 monotherapies (Figure 6e).

## 3. Discussion

Lysosomal degradation is a resistance mechanism for several intracellular acting anticancer therapeutics of different chemical and pharmacologic classes [39,40]. Drugs may accumulate in lysosomes through several ways including different forms of endocytosis [39], transporter-mediated uptake [6] and lysosomal sequestration of hydrophobic weak bases [12]. Within the lysosomes the drugs are inhibited to interact with their cytosolic target [41]. PCI is designed to overcome lysosomal entrapment of anticancer therapeutics, and the method has been shown to potentiate the activity of a variety of drugs including protein toxins, RNA, DNA, nanoparticles and also some conventional chemotherapeutic drugs [13,14]. This is the first report demonstrating direct mechanistic evidence of endo/lysosomal membrane localization and light-controlled cytosolic release of a smallmolecule inhibitor sequestered in endo/lysosomal compartments. Enhancement of sunitinib toxicity with PCI was, however, highly dependent on the treatment protocol, as no increase in cytotoxicity was observed when sunitinib was administrated prior to the photochemical treatment (“light after” strategy, Figure 1c,d). Using super-resolution fluorescence microscopy we were able to show subcellular co-localization of sunitinib and TPCS_2a_ (Figure 1f)_._ Our data indicate that sunitinib is photodamaged at pH 7 (Figure 1g), which is in agreement with Ohtsuki et al. who report increased endosomal pH prior to release of endocytosed molecules [27]. The absorption spectra (Appendix A) suggest that photodamage of sunitinib by TPCS_2a_ can also occur at pH 5. However, due to the amphiphilic nature of TPCS_2a_ [42], the PS will localize and accumulate in the cellular membranes in an in vitro and in vivo setting. This is in agreement with our SIM microscopy images (Figure 1f). Hence, the very close vicinity of these two drugs facilitates ROS-mediated photodamage of sunitinib after light-activation of TPCS_2a_ (Figure 1f,g). The light-triggered, ROS-induced damage of sunitinib most likely also took place when sunitinib-PCI was applied to the sunitinib resistant HT-29/SR cells. PCI was therefore unlikely to release and potentiate sunitinib that was accumulated in the in endo/lysosomal compartments in HT-29/SR cells. This was confirmed by the similar response to photochemical treatment in HT-29 and HT-29/SR cells, indicating no enhanced toxicity from cytosolic release of sunitinib in HT-29/SR cells (Figure 3i). Thus, ROS-mediated photodamage of sunitinib is probably one reason why the photochemical treatment applied here failed to potentiate lysosomal sequestrated sunitinib in resistant cells.

As observed in the parental HT-29 cells, sunitinib-PCI with the “light first” protocol was also found to increase the cell toxicity in HT-29/SR cells. This effect was however found to be additive at the highest concentration tested, indicating that the mechanism of sunitinib resistance is not related to lysosomal sequestration. Lysosomal sequestering of sunitinib has been associated with resistance by others [7,29,31]. To our knowledge, direct experiments addressing the cytotoxic impact of lysosomal accumulation of sunitinib in resistant cells has only been included in two reports where only modest increase in sunitinib toxicity was reported by combining sunitinib with Leu-Leu-O-Methyl in vitro [29] or chloroquine in vivo in sunitinib-resistant HT-29 xenografts [30]. These reports therefore also indicate other mechanisms than lysosomal sequestering to orchestrate sunitinib resistance. Further analysis on molecular pathways controlling cellular sunitinib-resistance is needed to conclude on the mechanisms of sunitinib-resistance in the HT-29/SR cells. Even though PCI failed to activate lysosomal sequestrated sunitinib in resistant cells, we here show that the technology can be applied to circumvent sunitinib resistance when combined with the protein toxin gelonin. PCI of protein toxins has previously been shown to circumvent resistance mediated through increased expression of P-glycoprotein [43,44] and the present results in the HT-29/SR cells further documents this strategy as highly efficient for the treatment of therapy resistant cancer.

Even though sunitinib-PCI with the “light first” protocol was shown highly efficient in vitro, the treatment did not result in synergistic tumor growth delay in mice. Continuous administration of sunitinib has previously been reported to induce a strong tumor growth delay in similar in vivo models [30,45]. Here, a moderate dose of sunitinib (two or six doses only) and TPCS_2a_ + light was selected to reach a therapeutic window for synergy evaluations. The lack of an increased treatment response with sunitinib-PCI may be due to a narrow therapeutic window. On the other hand, the in vitro data suggest PCI to potentiate sunitinib induced cytotoxicity at low sunitinib concentrations and doses of light. The “light first” protocol has been shown most efficient within ~2–3 h after the photochemical treatment [18], while the estimated time for maximum tumor accumulated sunitinib post-oral administration is 3–4 h [33]. There is therefore a possibility that the amount of sunitinib was too low to synergize with PCI. The lack of overall treatment response to sunitinib-PCI in vivo may also be a result of vascular responses. The photochemical treatment alone targets the tumor vasculature as shown by the IHC on CD31 in CT26.WT tumors in agreement with previous studies [37,38]. This vascular response may inhibit sunitinib penetration into the tumor when administrated after light exposure, and thereby counteract the action of sunitinib-PCI. The H&E data and IHC on Ki-67 supplied here demonstrate, however, larger treatment-induced necrosis in CT26.WT tumor bearing thymic animals receiving sunitinib-PCI compared to any of the control groups. This indicates that sunitinib had access to tumor after the photochemical treatment and induced an additive or supra-additive effect that is not reflected in the tumor growth measurements. The lack of overall treatment response following sunitinib-PCI despite the apparent larger treatment-induced necrosis may reflect a balance between growth of viable cells in the tumor rim and rate of removal of necrotic tissue.

Resistance towards vascular disruptive agents, including PS and light, has previously been associated with a residual tumor rim [46,47]. The tumors studied here by H&E and IHC were harvested several days after treatment when the tumors reached their endpoints of 1000 mm^3^. The absence of distinct treatment-induced viable rims was therefore not unexpected, and may have been present at earlier time points. Enhanced efficacy of sunitinib-PCI may therefore be expected by adjuvant treatment targeting this viable rim.

The treatment response following sunitinib-PCI in HT-29-bearing athymic mice was smaller than expected, and the response was even further reduced in CT26.WT-bearing immunocompetent mice (Figure 5a,b). PCI in combination with several different drugs has been shown to generate immune-mediated anticancer activity and the overall response has in general been better in immunocompetent mice compared to athymic mice [34,35]. Since the tumor models in athymic and immunocompetent mice here are of different cell line origin, the two models cannot be directly compared. Nevertheless, the IHC data demonstrate an antagonistic effect on CD3+ tumor infiltrating cells in CT26.WT tumors following sunitinib-PCI compared to the agonistic effect observed post- sunitinib- or TPCS_2a_ + light- monotherapy. The combination of sunitinib with immunotherapeutic approaches is somewhat controversial. Sunitinib has been shown to enhance intratumoral infiltation of CD8+ T-cells in combination with a CD40 agonist, a member of the tumor necrosis factor (TNF) superfamily [48] and to suppress immune regulatory cells in combination with celecoxib [49]. On the contrary, sunitinib has been reported to inhibit anti-tumor vaccination by decreasing antigen presenting cells [50] and to impair proliferation and function of phytohemagglutinin (PHA) stimulated human T-cells [51], in agreement with our results showing a decrease in tumor infiltrating T-cells following sunitinib-PCI. Jaini et al. also showed that the decreased immunoresponse following sunitinib and tumor vaccination could be avoided by careful scheduling of the applied drugs, and that enhanced vaccination could be achieved by administration of sunitinib after the priming phase of the vaccination [50]. It is therefore possible that the combination of sunitinib and TPCS_2a_+light would be more effective if sunitinib was delivered one week post-photochemical treatment. This procedure will, however, probably not induce cytosolic release of sunitinib from endocytic vesicles since the time-frame between the photochemical treatment and sunitinib administration will be too long.

Sunitinib has previously been suggested as a photosensitizer for endo/lysosomal destruction and it was argued that this approach could be used for the release of sunitinib entrapped in endo/lysosomal compartments [22]. As the activity of sunitinib is here shown to be reduced upon blue light exposure (Figure 1g), dual utilization of sunitinib as a TKI and a photosensitizer is probably little effective at the cellular level. The enhanced tumor growth delay observed by this approach may be due to pharmacologic interactions in the tumor and vascular system. The clinical potential of this suggestion is, however, highly limited due to the absorption maximum of sunitinib in the blue region where the tissue light penetration is only a few 100 µm into the tissue. 

## 4. Materials and Methods

### 4.1. Cell Lines and Cultivation

The human colorectal adenocarcinoma cell line HT-29 (ATCC HT-38™) and the murine colorectal carcinoma cell line CT26.WT (ATCC CRL-2638™) were obtained from the American Type Culture Collection (ATCC, Manassas, VA, USA). The cells were subcultured 2–3 times a week in McCoy’s 5a medium (Sigma-Aldrich, St. Louis, MO, USA) and RPMI-1640 (Sigma-Aldrich), respectively. The culture media were supplied with 10% fetal bovine serum (FBS) (Thermo Fisher Scientific, Waltham, MA, USA), 100 U/mL penicillin and 100 U/mL streptomycin (Sigma-Aldrich). Sunitinib-resistant HT-29 cells, named HT-29/SR, were generated by continuous exposure to 2 µM sunitinib for up to 5 months and the resistance was routinely verified during this time frame. Untreated parental HT-29 cells were cultured in parallel with the HT-29/SR cells. The cell lines were maintained at 37 °C in a humidified atmosphere containing 5% CO_2_.

### 4.2. Drugs and Chemicals

The photosensitizer (PS) TPCS_2a_ (PCI Biotech AS, Oslo, Norway) was dissolved at 0.4 mg/mL in 3% Tween 80, 2.8% mannitol, 50 mM Tris, pH 8.5 (all from Sigma-Aldrich), and kept protected from light at 4 °C. Sunitinib malate (PZ0012, Sigma-Aldrich) was for in vitro experiments dissolved in dimetylsulfoxid (DMSO) to a final concentration of 2.5 mM, stored as aliquots at −20 °C, and subjected to maximum two freeze-thaw cycles. Sunitinib malate for in vivo experiments was dissolved at a concentration of 8 mg/mL (20 mM) in a vehicle containing distilled water with 1.8% NaCl, 0.5% carboxymethylcellulose, 0.4% Tween 80 and 0.9% benzylalcohol. The pH of the solution was adjusted to 6.0. The sunitinib mixture was sonicated to achieve stable dispersion [45] and used within 24 h after sonication. All sunitinib reagents were stored protected from light. Recombinant gelonin (rGel) was generously provided by Dr. Michael Rosenblum’s laboratory at M.D. Anderson Cancer Center (Houston, TX, USA). Aliquots of rGel were stored in phosphate-buffered saline (PBS) at −20 °C. All experiments in vitro and in vivo with sunitinib and/or TPCS_2a_ were performed under subdued light.

### 4.3. In Vitro Light Sources

Illumination of the cells was performed with an in-house made diode lamp or a LumiSource™ lamp (PCI Biotech AS). The diode lamp delivers red light (E_max_ = 650–652 nm) at a fluence rate of 6 mW/cm^2^. LumiSource™ consists of four 18-W Osram L 18/67 light tubes and delivers blue light (λ= 400–500, λ_max_ = 435 nm) at a fluence rate of 7.1–9.6 mW/cm^2^.

### 4.4. In Vitro PCI Treatment

4000 HT-29 or HT-29/SR cells/well or 1500 CT26.WT cells/well were seeded in 96-well plates (Nuncon Delta, Thermo Fisher Scientific) and allowed to attach overnight. For clonogenic assay, 500 cells were seeded in 6-well plates (Nuncon Delta, Thermo Fisher Scientific). In the “light after” protocol, the cells were incubated with sunitinib prior to light exposure. The cells were incubated with sunitinib for 48 h, at indicated concentrations, prior to an 18 h co-incubation with 0.4 µg/mL TPCS_2a_. The cells were incubated in total 66 h with sunitinib. At the end of the incubation, cells were washed with PBS twice and chased 4 h in drug-free mediums before light exposure in order to remove plasma membrane bound-TPCS_2a_. In the “light first” protocol, the cells were subjected to light before sunitinib incubation. The cells were incubated with 0.4 µg/mL TPCS_2a_ for 18 h, washed with PBS twice and chased in medium for 4 h before light exposure. Sunitinib at indicated concentrations was then added immediately after illumination and incubated for 72 h. For the MTT-assay, sunitinib was removed and treatment efficacy was assessed as described in Section 4.5. For the clonogenic assay, the sunitinib incubation was sustained until the end of the experiment. PCI of rGel was only performed with the PCI “light after” protocol. Here, rGel at indicated concentrations was added to the wells during the chase period (4 h) and replaced with fresh medium before light exposure.

### 4.5. Treatment Efficacy In Vitro; Colony Formation Capacity, Viability and Proliferation

Cell viability was assessed using the MTT (3-(4,5-dimethyl-2-thiazolyl)-2,5-diphenyltetrazolium bromide). MTT was evaluated 48 h post-light exposure in all PCI “light after” protocols, except from experiments in Figure 1c. Experiments using PCI “light first” protocol was evaluated using the MTT assay 72 h post-light. Cells were incubated with 0.25 mg/mL MTT (Sigma-Aldrich) for 3–4 h. The medium was then removed, and the formazan-crystals were solubilized in DMSO. Absorbance was measured at 570 nm using a plate reader (PowerWave XS2 microplate spectrophotometer) with the Gen5 software program (Biotek Instruments Inc., Winooski, VT, USA). The colony formation assay was performed 10–14 d post-treatment, when sufficiently large colonies (>50 cells/colony) were formed in controls [52]. The medium was removed, and the cells were washed with 0.9% NaCl solution and fixed with absolute ethanol for 10 min. The colonies were stained with methylene blue (Sigma-Aldrich, 12 mg/mL in 0.1% NaOH) for 5 min. Colonies were manually counted under a magnifying glass using an automatic E-Count™ colony counter pen (Heathrow Scientific, Vernon Hills, IL, USA).

Real-time monitoring of cell proliferation was measured using IncuCyte FLR kinetic imaging system (Essen Bioscience, MI, USA). 4000 HT-29 or HT-29/SR cells/well were seeded out in 96-well plates. The cells were treated as indicated and monitored for up to a week. Phase-contrast images were acquired every 3 h in each well, processed using by IncuCyte software Rev2 and presented as cell confluence over time.

### 4.6. Intracellular Localization of TPCS_2a_ and Sunitinib by Fluorescence Microscopy

HT-29 or HT-29/SR cells were seeded on cover slips (No. 1014/10, Assistent, Sondheim, Germany) in 48-well plates (3 × 10^4^ cells/well) and allowed to attach overnight. The cells were then incubated with 0.4 μg/mL TPCS_2a_ for 18 h and either studied directly after the end of incubation or after wash and a 4-h chase in drug-free medium to remove TPCS_2a_ from the plasma membrane as done in the PCI procedures. For intracellular detection of sunitinib, attached cells were incubated for 24 h with 2 µM sunitinib. HT-29/SR cells were seeded out in the presence of 2 µM sunitinib to maintain sunitinib accumulation in the cells.

For evaluation of cytosolic release of sunitinib after PCI “light first” and “light after” procedure, 1.5 × 10^4^ HT-29 cells/well were seeded, allowed to attach and treated as described in Section 4.4 to mimic the PCI protocols. For the “light after” procedure, cells were incubated with 2 µM sunitinib for 48 h, and co-incubated with 0.4 µg/mL TPCS_2a_ for 18 h. The cells were washed and chased 4 h in drug-free medium before light exposure. In the “light first” procedure, the cells were incubated with 0.4 µg/mL TPCS_2a_ for 18 h, washed and chased 4 h before light exposure. Sunitinib, 6 µM, was added immediately after light exposure. Image acquisition after light exposure was performed at least an hour after illumination to allow cytosolic release of the endo/lysosomal content.

Lysosomes and endosomes were visualized by 30 min incubation with LysoTracker Red DND-99 or LysoTracker Green DND-26 (both from Life Technologies, Carlsbad, CA, USA) at 75 nM. Hoechst 33342 (Sigma-Aldrich) was added at 10 µM and incubated for 15 min to visualize the nucleus. Upon image acquisition, the cover slip was carefully removed from the well, washed twice in ice cold PBS with Ca^2+^ and Mg^2+,^ and inverted on a microscope slide. Image acquisition was performed with a Zeiss Axioplan epifluorescence and phase contrast microscope using 63x/NA1.4 PlanApo objective (Carl Zeiss AG, Oberkochen, Germany). The images were acquired with a cooled charge-coupled device (CCD) camera (AxioCam MRm camera, Carl Zeiss AG). The TPCS_2a_ fluorescence was recorded using a 395–440 nm excitation filter, a 460 nm dichroic mirror and a 620 nm long pass filter. For recording fluorescence from sunitinib or LysoTracker Green, a 450–490 nm band pass excitation filter, a 495 nm dichroic mirror and a 500–550 nm band pass emission filter was used. The LysoTracker Red fluorescence was recorded using a 595 nm excitation filter and a 620 nm emission filter. The AxioVision Analysis (Carl Zeiss AG) software program was used to process and analyze the images.

### 4.7. Subcellular Localization of Sunitinib and TPCS_2a_

Wide-field and structured illumination microscopy (SIM) were performed in order to determine the subcellular localization of sunitinib and TPCS_2a_ within the endocytic vesicles in live cells. 3 × 10^5^ HT-29/SR cells were seeded in glass bottom dishes (Cat. no. P35GC-1.5-10-C, MatTek Corporation, Ashland, MA, USA) and allowed to attach overnight. The cells were co-incubated with 2 µM sunitinib and 0.4 µg/mL TPCS_2a_ for 18 h, washed twice with PBS and chased in serum-free FluoreBrite DMEM (Thermo Fisher Scientific). Trolox (Sigma-Aldrich) was added to the cells 15 min prior to image acquisition. SIM imaging was performed on a DeltaVision OMX V4 Blaze 3D-SIM microscope (GE Healthcare, Chicago, IL, USA) equipped with an Olympus 60× 1.42 NA Plan Apochromat objective and sCMOS cameras. Sunitinib and TPCS_2a_ were excited with a 488 nm and a 647 nm laser, respectively, and imaged sequentially. Z-stacks were recorded with a z-spacing of 125 nm. For each focal plane, 15 raw images (five phases for three different angular orientations of the illumination pattern) were captured. The fluorescence was detected through the band-pass filters 528/48 nm for sunitinib and 683/40 nm for TPCS_2a_. SI-images were reconstructed and aligned using Softworx software (GE Healthcare), and further processed using ImageJ (https://www.nature.com/articles/nmeth.2089).

### 4.8. Cellular Accumulation of Sunitinib

For quantification of sunitinib accumulation in the HT-29 parental and sunitinib resistant cell line, 1.5 × 10^5^ cells per well were seeded in 6-well plates. The HT-29/SR cells were seeded in the presence of 2 µM sunitinib, whereas the parental cells where allowed to attach overnight in drug-free medium before adding sunitinib. Both HT-29 and HT-29/SR were then incubated with 2 µM sunitinib, and at the end of a 72 hour-incubation, the cells were detached with 0.25% (*w/v*) trypsin-0.53 mM EDTA (Sigma-Aldrich), washed with PBS and filtered through a 5 mL round-bottom tube with a cell strainer cap (Becton, Dickinson and Company, Franklin Lakes, NJ, USA). Sunitinib accumulation in cells was quantified using a BD LSRII flow cytometer (Becton, Dickinson and Company). Live and single cells were gated based on forward (FSC) and side scatter (SSC) parameters. Sunitinib was excited by a 100 mW 406 nm laser. The fluorescence was collected through a 585/42 nm band pass filter combined with a 545 nm long pass dichroic filter. Data were processed by the FlowJo version 10 software (Tree Star Inc., Ashland, OR, USA).

### 4.9. Absorption and Fluorescence Spectroscopy of TPCS_2a_ and Sunitinib

Photochemical damage of sunitinib in solution pH 7 was evaluated by absorbance and emission measurements. Sunitinib (0.15 µM) and TPCS_2a_ (0.15 µg/mL) were prepared in PBS without Ca^2+^ and Mg^2+^ (Sigma-Aldrich) and subjected to blue light exposure (LumiSource™). The sunitinib and TPCS_2a_ concentration was selected based on optical density <0.1 when both compounds are combined to avoid inner filter effect. The samples were illuminated in quartz cuvettes with lid, and sealed with Parafilm^®^ (Thermo Fisher Scientific) to avoid evaporation during illumination. The absorption and emission spectra were recorded immediately after light exposure at ambient temperature. Absorption spectra were recorded using a Shimadzu UV-2550 spectrophotometer connected to a computer with the software program UVProbe 2.62 (Shimadzu Corporation, Kyoto, Japan). All absorption spectra were recorded from 300 to 750 nm. Emission was recorded with a Cary Eclipse spectrofluorimeter (Agilent, Santa Clara, CA, USA) using the Scan Software V1.1. (Agilent) in the range 360–750 nm. The excitation and emission slit widths were 5 nm for TPCS_2a_ and 20 nm for sunitinib. TPCS_2a_ was excited at 420 nm and sunitinib at 432 nm, based on the recorded absorption spectra (Appendix A). Relative decrease in peak intensity was calculated at 656 nm and 505 nm for TPCS_2a_ and sunitinib, respectively. Absorption spectra was also collected at pH 5 where sunitinib was prepared in citrate-phosphate buffer containing 1% FBS (Thermo Fischer Scientific). This citrate-phosphate buffer was prepared by dissolving sodium citrate tribasic dehydrate (Sigma-Aldrich) and disodium hydrogen phosphate dehydrate (Merck KGaA, Darmstadt, Germany) in distilled water [53] and pH-adjusted to pH 5.

### 4.10. Animals

All animal procedures were performed according to protocols approved by the Norwegian Food Safety Authority (FOTS ID 4593), which is the national animal research authority and were conducted according to the regulations of the Federation of European Laboratory Animal Science Association (FELASA). Handling of animals was therefore performed in compliance with EUs Directive 2010/63/EU on the protection of animals used for scientific purposes. Two different strains of female mice were used in this study. HSD athymic Nude-Foxn1^nu^ mice were bred at the Department of Comparative Medicine at the Norwegian Radium Hospital, Oslo University Hospital. BALB/c mice were obtained from Envigo (Horst, The Netherlands). The mice were maintained under specific pathogen-free conditions in a temperature-controlled room. Food and water were supplied ad libitum. The mice were on average 17–20 g (4–8 weeks old) when experiments were initiated.

### 4.11. Tumor Grafts

HT-29 cells (2.5 × 10^6^) in 30 µL PBS or CT26.WT cells (1 × 10^5^) in 15 µL PBS were injected subcutaneously on the left flank in athymic Nude-Foxn1^nu^ or thymic BALB/c mice respectively. The tumors and body weight were monitored 2–3 times per week. Tumor size was calculated using the following formula: V = (W^2^ × L)/2, where W is the width and L the length of the tumor measured by a digital caliper. The protocol was designed with two endpoints; tumor size 1000 mm^3^ and weight loss ≥20%. The animals were euthanized by cervical dislocation.

### 4.12. In Vivo Experimental Design and Methods

TPCS_2a_ was administered intravenously through the lateral tail vein at 5 mg/kg five days (CT26.WT) or seven days (HT-29) after tumor inoculation. Seventy-two hours post-TPCS_2a_ administration, the tumors were irradiated using a 652 nm red diode laser (CeramOptec GmbH, Bonn, Germany) at an irradiance of 90 mW/cm^2^ and a total dose of 15 J/cm^2^ for Nude-Foxn1^nu^ and 10 J/cm2 for BALB/c. The mice were anesthetized (sevofluran inhalation) and kept on a 37 °C heating pad during light exposure. The penetration depth of 652 nm red light into tissue is 4–5 mm [54] which is sufficient for the s.c. tumors at ~100 mm^3^ here subjected to light exposure. The selected light doses are in addition previously reported as sufficient for PCI [35]. Before light exposure, the animals were covered with aluminum foil with an opening diameter 2–3 mm larger than the tumor. Two different procedures, one with 2 doses of sunitinib (Sun1) and the other with 6 doses of sunitinib (Sun2) were performed for assessment of therapeutic effects of sunitinib-PCI in vivo. In both procedures sunitinib (40 mg/kg) was administered by oral gavage [30,45] using a feeding needle (22 Gauge, 25mm, #7901, Angthos AB, Lidingö, Sweden) 3 h prior to [22] and 30 min after light exposure. In the Sun2 procedure administration of sunitinib at 40 mg/kg was continued for four more days, to a total of 6 administrations. Animals receiving TPCS_2a_ were kept in the dark for one week after administration of TPCS_2a_ or three days after sunitinib administration to avoid photo-toxicity. For evaluation of tumor growth delay, animals were randomized into six treatment groups including no treatment (NT), PS and light (PS + light), sunitinib only Sun1-procedure (Sun1), sunitinib only Sun2-procedure (Sun2), Sun1-PCI and Sun2-PCI. The experiment was set up with 3 treatment sessions for each tumor model and NT, PS + light and sunitinib only controls were included in each session.

Intratumoral distribution of TPCS_2a_ and sunitinib in vivo was only evaluated in HT-29-bearing athymic Nude-Foxn1^nu^ mice with the Sun1 protocol. Animals were randomized into three groups; no treatment, TPCS_2a_ + Sun1 and Sun1-PCI with two mice in each group. 30 min post-light exposure the animals were sacrificed and the tumors were immediately placed on liquid nitrogen. 8 µm freeze sections were prepared and images acquired using Zeiss microscope as described above with 5×, 20× and 40× magnification immersion objectives (Carl Zeiss AG). TPCS_2a_ and sunitinib fluorescence was recorded using the filter combinations described in Section 4.6. The AxioVision software program (AxioVs40, version 4.8.0.0, Carl Zeiss AG) was used to process and analyze the images.

For IHC, NT-, PS + light-, Sun2- and Sun2-PCI-treated CT26.WT tumors in BALB/c mice reaching 1000 mm^3^ were harvested, fixed in formalin and embedded in paraffin before they were prepared as previously described [37] using αCD3 (A0452, Dako, Agilent Technologies), αKi-67 (ab15580, Abcam, Cambridge, UK) and αCD31(ab28364, Abcam). The tumor sections (2.5–3 µm) were also stained with H&E for evaluation of viable/necrotic tumor regions. Images were acquired using an AxioImager Z1 CellObserver microscope system (Carl Zeiss AG) with a 20x/NA0.8 lens and a 1ccc1 CCD camera (Carl Zeiss AG). 6 tiles were imaged with 10% overlap using an automatic stage. The images were stitched together using the Zen blue software (Carl Zeiss AG). The imaged area was selected to visualize the tumor from the distal and into the central part. Three ROI was defined at distant sites of the imaged area avoiding necrotic tissue. For CD3 stains, the number of positive cells in each ROI was counted and presented as an average. Two tumors were evaluated from each treatment group except from the PCI-Sun2 group, where three tumors were analyzed. 

### 4.13. Evaluation of Combination Therapy and Statistical Analysis

Evaluation of synergy of sunitinib-PCI treatment was determined with a statistical model based on the assumption that PS + light and sunitinib have distinct and independent mechanisms of action [55,56]. The theoretical additive effect in this model is a product of the survival fraction (SF) of each treatment separately calculated as follows: SF_add_ = SF_sunitinib_ × SF_PS+light_ (or log SF_add_ = log SF_sunitinib_ + log SF_PS+light_). The calculated SF_add_ was compared to the observed combined effect (SF_comb_). Synergy was further evaluated using the parameter DL (difference in logarithm) between observed SF_comb_ and the calculated SF_add_. DL = −(log SF_comb_ − log SF_add_) = log SF_add_/log SF_comb_ = log SF_sunitinib_ + log SF_PS+light_ − log SF_comb_. Synergistic effects resulted in positive DL values, antagonistic effects resulted in negative values and additive effects close to zero. Significant deviation from zero was established through one sample *t*-tests. Sigmaplot version 14.0 (Systat Sofware Inc., San Jose, CA, USA) was used for statistical analysis where *p* ≤ 0.05 was considered statistically significant. Two-sided student’s *t*-test was performed for in vitro data, unless otherwise stated. For in vivo experiments, one-way ANOVA test and Holm-Sidak post-hoc test were performed to evaluate significant differences in tumor growth between the treatment groups. Statistical differences in survival were evaluated by pairwise log-rank analysis in IBM SPSS Statistics version 25.0 (IBM, Armonk, NY, USA).

## 5. Conclusions

In conclusion, this is the first report demonstrating cytosolic delivery of a small-moleculeinhibitor by PCI. Sunitinib-PCI was found highly promising in vitro when utilizing the “light first” protocol and was also indicated to increase treatment-induced necrosis in vivo. The overall treatment response in our animal models was, however, less than expected which indicate mechanisms in the tumor stroma to attenuate an overall treatment response. Particularly in CT26.WT tumor-bearing thymic animals where an antagonistic effect on infiltrating T-cells was observed. Hence, our work indicates PCI to potentiate sunitinib cytotoxicity although adjuvant therapy aimed at the tumor stroma should be evaluated to improve the therapeutic efficacy.

## Figures and Tables

**Figure 1 cancers-12-00417-f001:**
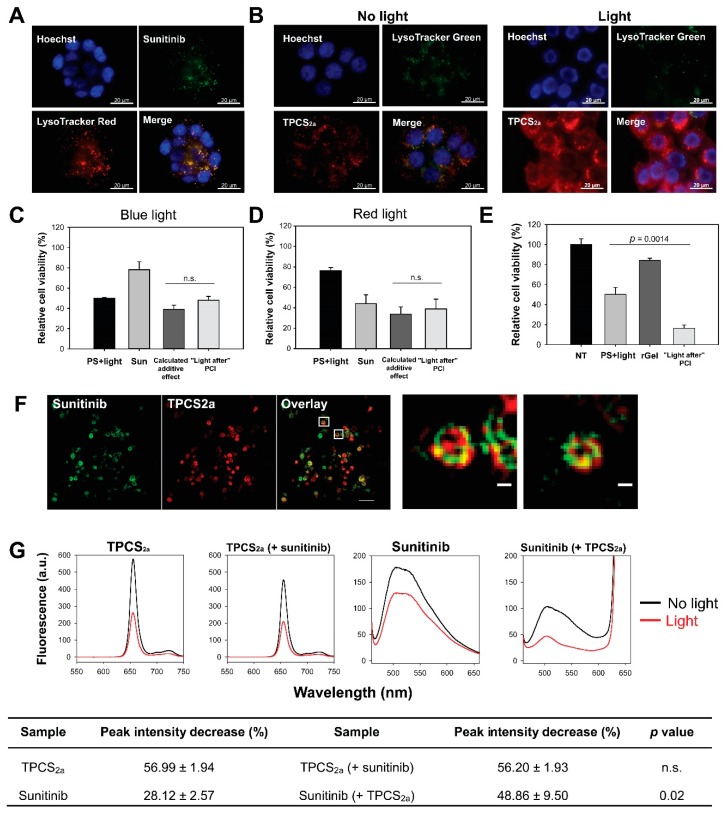
Close proximity of photosensitizer and sunitinib in endo/lysosomal membranes results in photochemical damage of sunitinib and lack of enhanced cytotoxicity with “light after” sunitinib-photochemical internalization (PCI). Representative fluorescence microscopy images of (**a**) intracellular co-localization (yellow) of sunitinib (green) and LysoTracker Red (red) after 24 h 2 µM sunitinib incubation in live HT-29 cells and (**b**) LysoTracker Green (green) and TPCS_2a_ (red) co-localization (yellow) after 18 h 0.4 µg/mL TPCS_2a_ incubation and 4 h wash (left) followed by 60 s blue light exposure (right). Blue: Hoechst 33342 stained nucleus. Scale bars: 20 µm. Cellular viability (MTT) of HT-29 cells (**c**) post-PCI “light after” procedure of 1 µM sunitinib (48 + 18 h incubation) with blue light at LD_50_ (~60 s) or (**d**) post-PCI “light after” procedure of 8 µM sunitinib (48 + 18 h incubation) with 90 s red light (mean of three experiments ± S.E.) or (**e**) 0.5 µM rGel using 60 s blue light (representative experiment of three, mean of triplicates ± S.D.). 60 s blue light ≈ 0.58 J/cm^2^, 90 s red light ≈ 0.54 J/cm^2^. (**f**) Superresolution (structured illumination microscopy, SIM) images of 2 µM sunitinib (green) and 0.4 µg/mL TPCS_2a_ (red) in live HT-29/SR cells after 18 h TPCS_2a_ incubation and 4 h chase. Co-localization indicated in yellow. Images are presented with maximum intensity projection of seven z-sections. One single z-section is presented for the enlarged images. Scale bars: 2 µm and 200 nm (enlarged) (**g**) Representative fluorescence emission spectra of 0.15 µg/mL TPCS_2a_, 1.5 µM sunitinib, and the combination in phosphate-buffered saline (PBS) with 1% fetal bovine serum (FBS) before and after blue light exposure (≈18.9 J/cm^2^) at pH 7. Data in the table are presented as decrease in peak intensity (%) after light exposure (mean of three experiments ± S.E.). n.s.: not significant. Statistical significance calculated with Student’s test (two-tailed *p* value).

**Figure 2 cancers-12-00417-f002:**
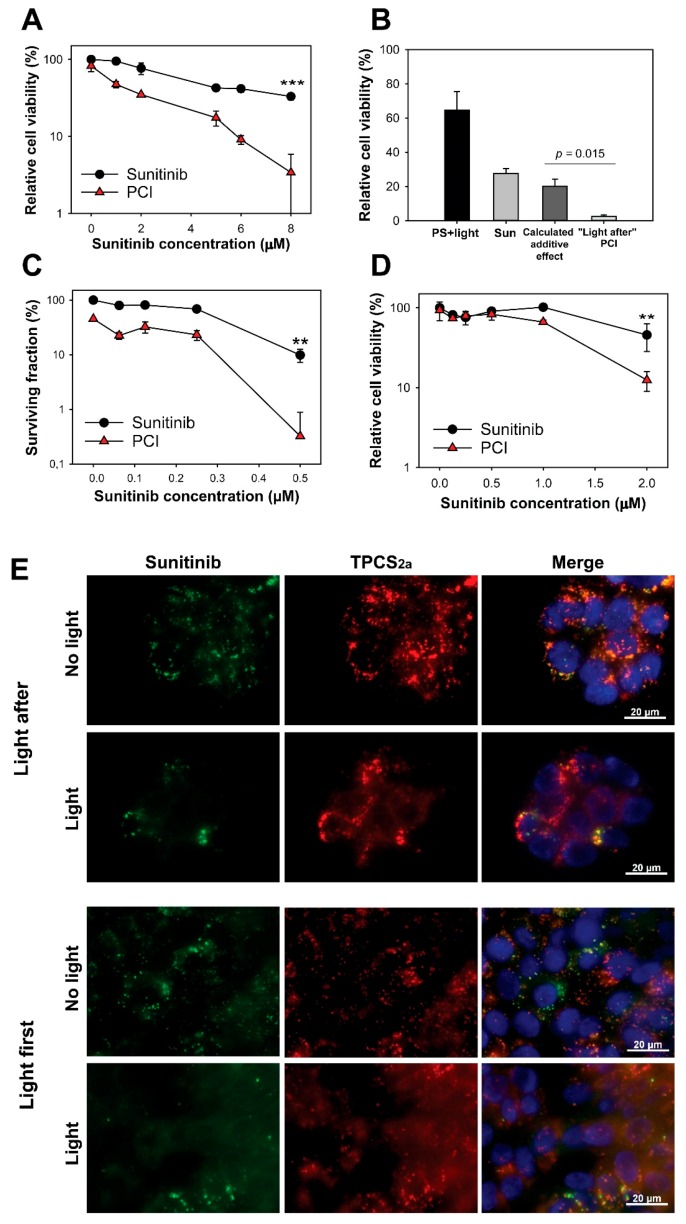
“Light first” sunitinib-PCI induces a synergistic cytotoxic treatment response. Cellular viability (MTT) of HT-29 cells post-PCI “light first” of (**a**) sunitinib at increasing concentrations (representative of three experiments, mean of triplicates ± S.D.) or (**b**) 8 µM sunitinib (data are mean of three independent experiments ± S.E.) exposed to 90 s red light. (**c**) PCI “light first” of sunitinib at increasing concentrations evaluated with clonogenic assay (60 s blue light, representative experiment of three, mean of triplicates ± S.D.). (**d**) Cellular viability (MTT) of CT26.WT cells post-PCI “light first” of sunitinib exposed to 40 s blue light. Sun: sunitinib. Statistical significance calculated with Student’s test (two-tailed *p* value) where *** indicates *p* ≤ 0.001 and ** *p* ≤ 0.01. Cells were incubated with sunitinib for 72 h after light-exposure (**e**) Representative live cell fluorescence microscopy images of “light after” PCI of 2 µM sunitinib and (**f**) “light first” PCI of 6 µM sunitinib before and 1 h after blue light exposure (60 s). Sunitinib (green), TPCS_2a_ (red), Hoechst-stained nucleus (blue). Co-localization indicated in yellow. Scale bar: 20 µm. 60 s blue light ≈ 0.58 J/cm^2^, 90 s red light ≈ 0.54 J/cm^2.^

**Figure 3 cancers-12-00417-f003:**
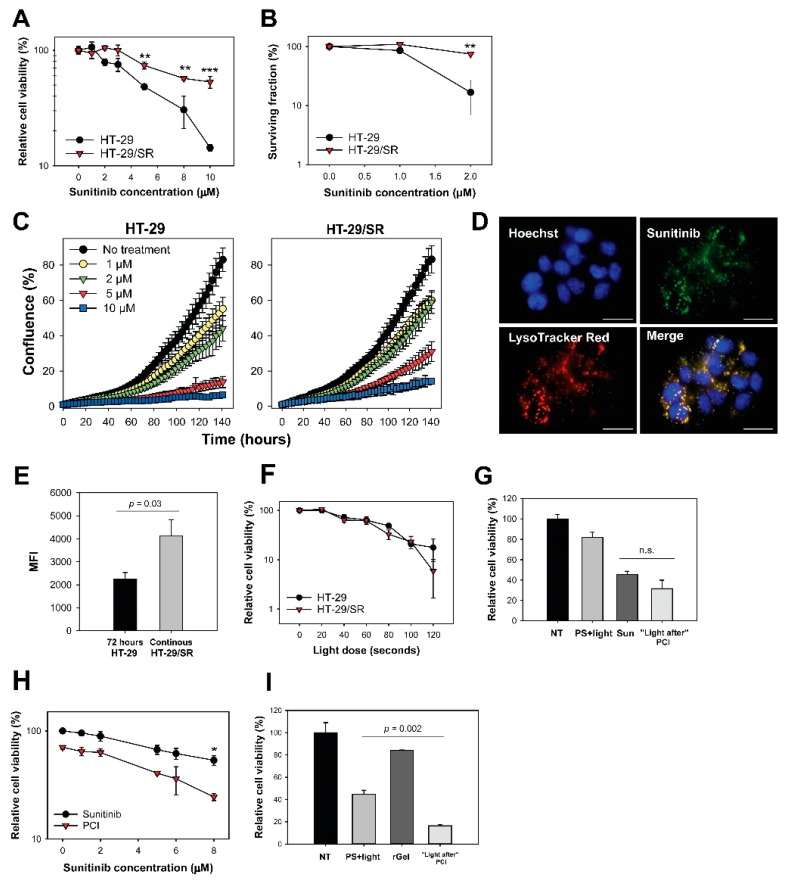
“Light first” sunitinib-PCI enhance cytotoxicity in HT-29/SR cells but cannot overcome sunitinib resistance. (**a**) Relative viability 72 h after sunitinib incubation measured by MTT. The HT-29/SR cells had been exposed to sunitinib for 3 months and were seeded in sunitinib-free medium. The graph is a representative experiment of five, mean of triplicates ± S.D. Sunitinib resistance in HT-29/SR cells was verified with (**b**) clonogenic assay and (**c**) proliferative capacity. The data points show % confluence at different sunitinib concentrations using IncuCyte live-cell analysis system. HT-29/SR cells were seeded out without sunitinib present. (**d**) Live cell fluorescence image of HT-29/SR showing co-localization (yellow) of sunitinib (green) and LysoTracker Red (red). Nucleus stained with Hoechst 33342 (blue). HT-29/SR cells were continuously incubated with sunitinib. Scale bar = 20 µm. (**e**) Evaluation of sunitinib accumulation in HT-29 (72 h incubation) and HT-29/SR (long-term sunitinib exposure) cells with flow cytometry. HT-29/SR cells were continuously incubated with sunitinib. Median sunitinib fluorescence intensities in live and single cells (mean of three experiments ± S.E.). (**f**) Photochemical treatment (photosensitizer and light) response of HT-29 and HT-29/SR evaluated with MTT assay post-90 s red light exposure (representative experiment of three, mean of triplicates ± S.D.). Cellular viability of sunitinib in HT-29/SR cells using (**g**) “light after” with 8 µM sunitinib or (**h**) “light first” protocol assessed by MTT post-90 seconds red light exposure, respectively (representative data based on three independent experiments, mean of triplicates ± S.D.). HT-29/SR cells were seeded in sunitinib-free medium. (**i**) Cell viability after PCI “light after” of 0.5 µM rGel as assessed by MTT post-60 seconds blue light exposure (representative experiment of three, mean of triplicates ± S.D.). 60 s blue light ≈ 0.58 J/cm^2^, 90 s red light ≈ 0.54 J/cm^2^ NT: no treatment, Sun: sunitinib. Statistical significance calculated with Student’s test (two-tailed *p* value) where *** indicates *p* ≤ 0.001, ** *p* ≤ 0.01 and * *p* ≤ 0.05, n.s.: not significant.

**Figure 4 cancers-12-00417-f004:**
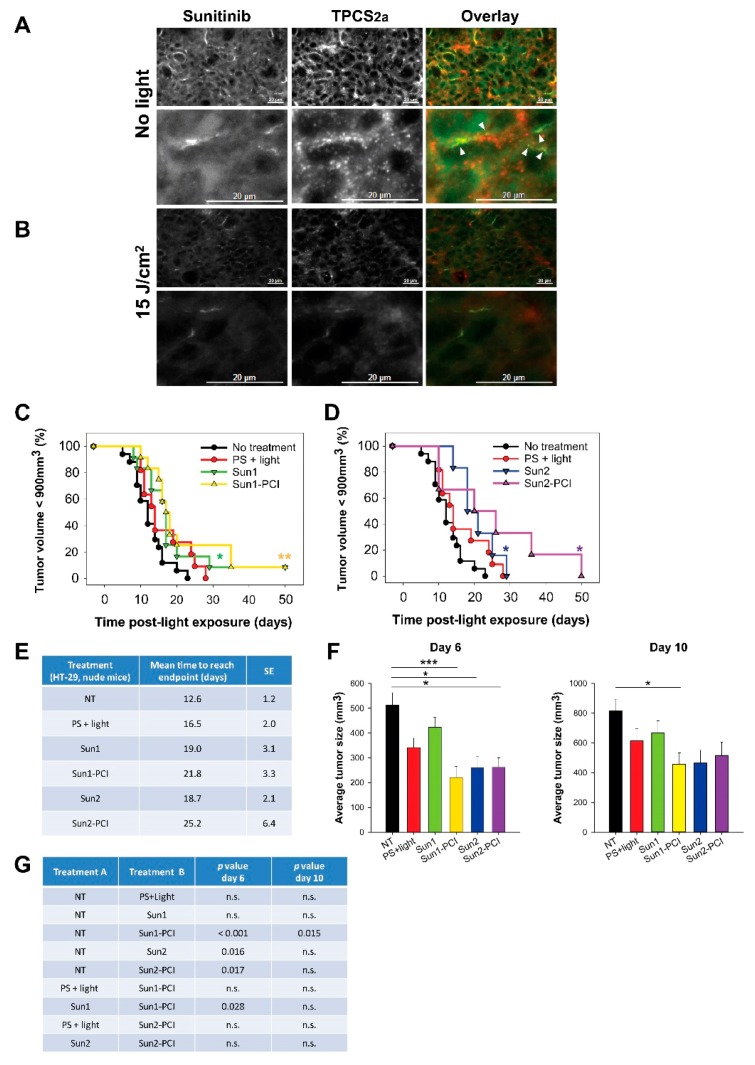
Sunitinib-PCI treatment response of HT-29 xenografts in athymic Nude-Foxn1^nu^ mice Intra-tumoral distribution of sunitinib (green) and TPCS_2a_ (red) (**a**) before and (**b**) 30 min post-light exposure of HT-29 tumors treated with the Sun1- PCI protocol. Error bars: 20 µm Co-localization (yellow) indicated with white arrows. Kaplan-Meier plots illustrating overall treatment response following (**c**) Sun1- and (**d**) Sun2-PCI, * indicates significance compared to no treatment. Statistical significance established by pairwise long-rank analysis. (**e**) Mean estimated time to reach endpoint in each treatment group. SE: standard error. (**f**) Average tumor size in the indicated treatment groups at day 6 (left) and day 10 (right) post-light exposure. Statistical significance with asterisk where *** indicates *p* ≤ 0.001 and * *p* ≤ 0.05. (**g**) Table of *p* values is shown in cases where the difference in tumor size between the treatment groups is significant (*p* ≤ 0.05) at day 6 and day 10. Significant difference established by one-way ANOVA test followed by pair wise multiple comparison procedure (Holm-Sidak). NT: No treatment, n.s.: not significant.

**Figure 5 cancers-12-00417-f005:**
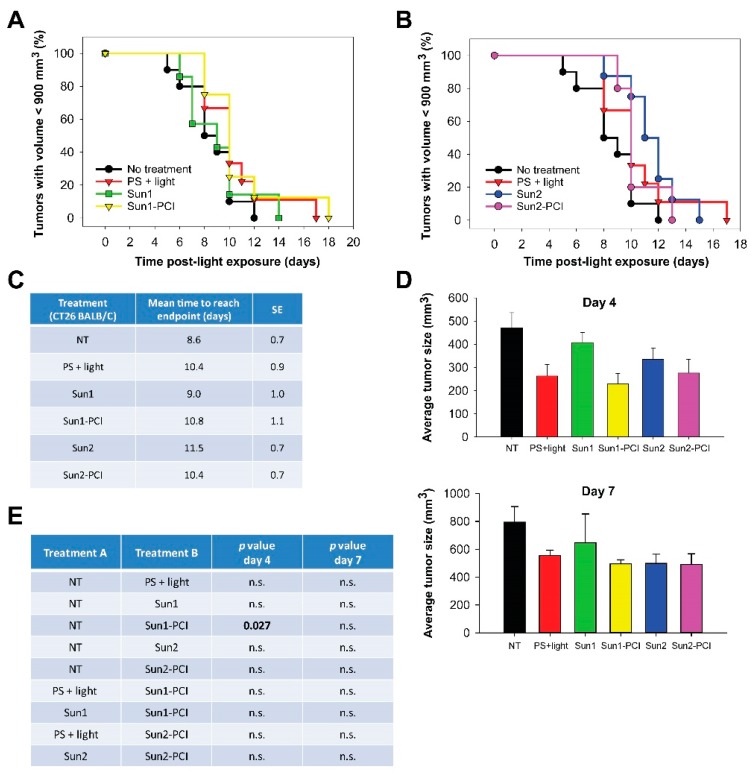
Sunitinib-PCI treatment response of CT26.WT allografts in BALB/c mice. Kaplan-Meier plots illustrating treatment responses following (**a**) Sun1- and (**b**) Sun2-PCI, where asterisk indicates significance compared to no treatment. (**c**) Mean estimated time to reach endpoint in each treatment group. (**d**) Average tumor size in the indicated treatment groups at day 4 (upper panel) and day 7 (lower panel) post-light exposure. (**e**) Table of *p* values is shown in cases where the difference in tumor size between the treatment groups is significant (*p* ≤ 0.05) at day 4 and day 7. Significant difference established by one-way ANOVA test followed by pairwise multiple comparison procedure (Holm-Sidak). NT: No treatment, n.s.: not significant.

**Figure 6 cancers-12-00417-f006:**
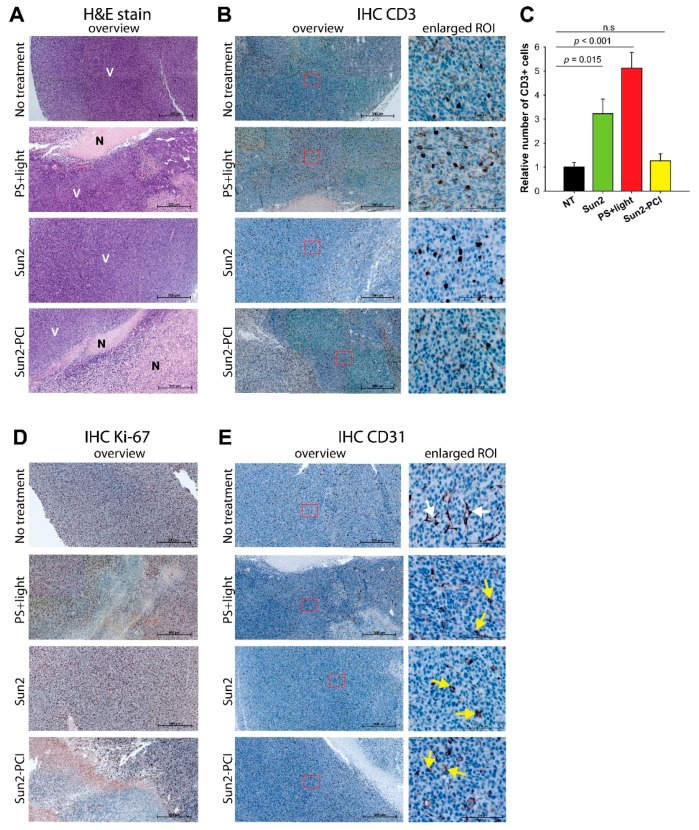
IHC of CT26. WT tumor tissue sections from BALB/c mice treated with the Sun-2-PCI procedure. Representative images of CT26.WT tumor sections, following PS + light, Sun2 or Sun2-PCI treatment. (**a**) H&E stain (N: necrotic V: viable) and (**b**) CD3 stain. (**c**) Quantification of CD3 staining based on three ROIs in each tumor (two tumors in each group). Mean ± S.E. Significant difference established by one-way ANOVA test followed by pair wise multiple comparison procedure. (**d**) Ki-67 stain and (**e**) CD31 stain. Arrows indicate intact (white) and collapsed (yellow) vessels. Magnification in overview 20×, scale bar: 500 µm. ROI: region of interest.

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
