# Peer review of "Photochemically-Induced Release of Lysosomal Sequestered Sunitinib: Obstacles for Therapeutic Efficacy"

_cancers, 2020, doi:10.3390/cancers12020417_

Round 1

Reviewer 1 Report

The paper entitled “Light-induced release of lysosomal sequestered sunitinib” by JJ Wong et al. is based on a good rationale even if the presented results are unfortunately not in line with author’s expectations. Paper organization is good and the effects of PCI on sunitinib release and cytotoxicity has been deeply investigated both in vitro both in in vivo colon cancer models. However, the major concerns derived from the choice and the description of the drug/light time adopted for combination therapy. The two “light protocols” used are very different, not only regarding TPCS2a activation but also for Sun incubation duration, being not only light the unique variable. In fact, the authors themselves concludes that therapy outcome is strictly related to the adopted treatment protocols. The in vitro results on enhanced Sun efficacy are not confirmed in animals, where tumor response seem to be not different between Sun and Sun+PCI. In any case the discussion of the presented data is well argued by analyzing punctually the several variables responsible for the “failure” of Sun-PCI treatment. Of particular interest is the analysis of tumor tissue response, especially regarding T-cell infiltration.

In any case a minor revision is requests to address the following questions and make the experimental description and data presentation more easily readable.

Abstract: sentences from line 22 to line 26 are quite confused and of poor meaning without reading the full paper. Paragraph 2.2. The nomenclature “light after” (and “light first”) is quite inappropriate and a short description of the protocol must be added in the text also when presenting the results, trying to well indicate drug incubation time and end-point for MTT assay measurements. No references are cited (nor in the experimental section) when talking about theoretical additive effect. Why the protocol of incubation and Sun doses have been changed to make the comparison between blue and red light? (Fig 1 c,d). In general, I suggest to the author to add a graphical overview of the protocols/drug ratio/incubation times used (in materials and methods or supplementary files) to help the reader. As an example, in paragraph 4.4 is not clear if for this experiment the cells are incubated with Sun for 48+18 h (18 h together with TPCS2a) or if Sun is incubated for a total of 48 h (30 h alone + 18 h with TPCS2a). Paragraph 4.4 have to be revised to explain more precisely incubation time adopted in the work. Fig 1 c-d-e: why cell viability is expressed as relative? Relative to controls for c and d, but in e also NT is presented. Fig 1 F: why SIM was assed in HT29/SR cells and not in wt cells? It can be observed a different localization depending on the extent of drug sensibility/resistance? Paragraph 4.9 and 2.3: is the ratio TPCS2a/Sun used for spectra acquisition comparable with those used in cell experiments? Paragraph 2.4: As in point 2, also for “light first” protocol more information has to be added also in the results sections. In general, the description as well as figure 2 (and relative caption) are quite confused. Also the experimental choice of the light source select for the different experiments presented in this paragraph is not clear. 2a: PCI in the legend means PCI+Sun. Please modify the legend accordingly. Please check the labeling of y axis.

Can be performed an analysis of the extent of synergism at the different Sun concentrations used? In the text is declared that “the reduction of cell viability is synergic”.

Sun concentration used for MTT assay and clonogenic assay are completely different. Please comment on this point. Why for CT26WT cells blue light is used? Which is the rationale to change the light source to demonstrated the same tendency already showed for HT29 cells? Also in in vivo experiments red light is used. Line 172-178: the results on fluorescence microscopy are not well explained as well as the incubation protocol (paragraph 4.6). Paragraph 2.5: how Sun resistance in HT-29/SR correlates with proliferative capacity? (Fig. 3 c). Please add some comments in the result section. 3e and caption: please explain better the significance of “continuous”. Indicate the exact incubation time. In general, the paragraph explaining Sun accumulation results is not well presented. Paragraph 2.5: line 213-215. Please show the data together with that shown in Fig. S4. Paragraph 2.7: basing on the completely different condition/environment/drug distribution between cells cultured in monolayers and xenografted in athymic mice, while HT29/SR cells have been excluded a-priori (rationale: no PCI effect on Sun resistance observed in vitro) for in vivo studies? Please comment on this. Paragraph 2.8, Fig 5 d. In the CT26.WT mice model the effect on tumor size of PCI+light is quite important, especially at day 4 and basically higher than that of Sun. TPCS2a acts as a pure photosensitizer rather than only as a drug involved in lysosome destabilization. Differently, in the in vitro studies using red light a maximum of 20-30% viability reduction was related to PS+ light. Please comment on this. Paragraph 4.4 and 4.5 must be revised to better explain “light after” and “light first” protocols, and probably jointed. Paragraph 4.6: why HT29/SR have been seeded in the presence of Sun? It is unlikely that the resistance can be lost during seeding time. Control all superscript characters in the text.

Author Response

Reviewer 1

We thank the reviewer for a thorough process which clearly has improved our manuscript

Abstract: sentences from line 22 to line 26 are quite confused and of poor meaning without reading the full paper.

In the revised manuscript we have changed these sentences as follows; "PCI also inhibited lysosomal sequestering of sunitinib in HT29/SR cells with acquired sunitinib resistance, but did not reverse the resistance. The mechanism of acquired sunitinib resistance in HT29/SR cells was therefore not related to lysosomal sequestering."

Paragraph 2.2. The nomenclature “light after” (and “light first”) is quite inappropriate and a short description of the protocol must be added in the text also when presenting the results, trying to well indicate drug incubation time and end-point for MTT assay measurements.

The terms "light after"- and "light first"-PCI reflects the two treatment protocols for PCI where the photochemical reaction is exerted either after (light after) or before (light first) administration of sunitinib. These terms were established already in 2002 when the first results on PCI with the "light first" protocol was published (Prasmickaite et al., British J Cancer 2002, ref. 18), and has later been documented in several publications (Weyergang et al., J Controlled Release 2006, Yip et al., Molecular Pharmaceutics 2006, Berstad et al. Biochimica et Biophysica Acta 2012). In the new version of the manuscript we have included a brief description of the light after procedure both in the last part of the introduction and also in 2.2.  

No references are cited (nor in the experimental section) when talking about theoretical additive effect.

We thank the reviewer for addressing this. The following reference is now included in the experimental section line 700 (Steel and Peckham, Int J Radiat Oncol Biol Phys, 1979, ref 55, and Selbo et al., Photochem Photobiol, 2001, ref 56). 

Why the protocol of incubation and Sun doses have been changed to make the comparison between blue and red light? (Fig 1 c,d). In general, I suggest to the author to add a graphical overview of the protocols/drug ratio/incubation times used (in materials and methods or supplementary files) to help the reader. As an example, in paragraph 4.4 is not clear if for this experiment the cells are incubated with Sun for 48+18 h (18 h together with TPCS2a) or if Sun is incubated for a total of 48 h (30 h alone + 18 h with TPCS2a). Paragraph 4.4 have to be revised to explain more precisely incubation time adopted in the work.

To avoid any confusion and make the revised manuscript more readable, we have included information on sunitinib incubation in the figure legends. Paragraph 4.4 has also been revised as suggested by the reviewer (line 526-534).

5.Fig 1 c-d-e: why cell viability is expressed as relative? Relative to controls for c and d, but in e also NT is presented.

The viability data are expressed as relative to non-treated (NT) cells throughout the manuscript. In the revised version, we have removed the NT bar from Figure 1e to avoid any confusion.

6.Fig 1 F: why SIM was assessed in HT29/SR cells and not in wt cells? It can be observed a different localization depending on the extent of drug sensibility/resistance?

The localization pattern of sunitinib to Lysotracker-positive compartments is in this manuscript observed in both HT-29 and HT29/SR cells (Figure 1a and figure 3a). The SIM experiments are highly time-consuming and have also a high cost, and we decided to proceed with SIM only in one of the cell lines (HT29/SR).

Paragraph 4.9 and 2.3: is the ratio TPCS2a/Sun used for spectra acquisition comparable with those used in cell experiments?

The ratio between TPCS2a and sunitinib in the cellular incubation media is comparable to that used for spectra acquisition (0.4 µg/ml TPCS2a and 1-8 µM sunitinib (Figure 2a) versus 0.15 µg/ml TPCS2a and 1.5 µM sunitinib (Figure 1g). However, we do not know how this ratio appear within the membrane of the endocytic vesicle where the reaction takes place.

Paragraph 2.4: As in point 2, also for “light first” protocol more information has to be added also in the results sections. In general, the description as well as figure 2 (and relative caption) are quite confused. Also the experimental choice of the light source select for the different experiments presented in this paragraph is not clear.

Here we first documented sunitinib to endocytic vesicles (Figure 1a). This localization indicated that PCI should be effective and we first tried with the general protocol (PCI light after) using blue light which is a standard light source for PCI in vitro. However, using this protocol in combination with blue light was not effective (Figure 1c). We thought the lack of PCI efficacy could be due to the blue light source which peaks in the same area as the sunitinib absorption spectrum (Supplementary Figure S3), and we therefore performed PCI (light after) with a red light source. This was, however, also ineffective (Figure 1d). We then hypothesized that PCI of sunitinib was not working due to damage by ROS (generated through activation of the photosensitizer). Our data on SIM (Figure 1f) and fluorescence/absorption spectra at pH 7 (Figure 1g) indicate that this is likely. PCI has previously also been documented as effective when the drug is given after the photochemical treatment (light first), and we decided to try the PCI light first procedure to avoid ROS-mediated damage of sunitinib which was successful in vitro (Figure 2).

We understand the reviewers comment that this manuscript includes several protocols for sunitinib-PCI. However, it is very much the changes in protocols throughout the manuscript that has led to our conclusions. We feel this is already described in detail throughout the result and discussion part of the manuscript, and we would like the editors opinion on whether this should be further explained. Please also find our comments to question 2 and 4.

8.2a: PCI in the legend means PCI+Sun. Please modify the legend accordingly. Please check the labeling of y axis.

In the revised manuscript we have changed "PCI" to Sun-PCI light after/light first to make it easier to follow. Regarding labeling of the y-axis we are unsure what should be changed.

Can be performed an analysis of the extent of synergism at the different Sun concentrations used? In the text is declared that “the reduction of cell viability is synergic”.

To calculate if sun-PCI is synergistic one must compare the observed effect with the calculated (theoretical) additive effect. The theoretical additive values are included in both Figure 1c-d and in Figure 2b. Such calculations are, however, not meaningful if one of the treatment is sub-toxic such as in 2a, 2d, 3a or 3b. On the other side, we here show a significant difference (t-test) between sunitinib and sunitinib-PCI at a photochemical treatment dose which do not induce cell death, indicating synergism on all tested concentrations.

10.Sun concentration used for MTT assay and clonogenic assay are completely different. Please comment on this point.

The sunitinib incubation in the clonogenic assay is much longer than the MTT assay and sunitinib has therefore a prolonged time to exert its mechanism of action. The sunitinib concentrations used in the clonogenic assay is therefore smaller than used for MTT. This explanation has now been included in paragraph 4.4 (line 533-534).

11.Why for CT26WT cells blue light is used? Which is the rationale to change the light source to demonstrated the same tendency already showed for HT29 cells? Also in in vivo experiments red light is used.

The blue light source is the standard light source for in vitro PCI in our lab. The penetration depth of blue light is up to 1 mm (Ash et al., Laser Med Sci, 2017, ref. 54), which is adequate for the purpose of examining the effect on cell monolayer. Since the Sunitinib-PCI efficacy seemed not to be related to the light source (red or blue) (Figure 1C and D) we selected the blue light source for this experiment. In vivo, a red laser is used due to the much higher light penetration in tissue. Please refer to line 486-488 and 659-660 in the manuscript regarding choice of light source and tissue penetration in vivo. Please also find our response to question 7.

12.Line 172-178: the results on fluorescence microscopy are not well explained as well as the incubation protocol (paragraph 4.6).

In the new version of our manuscript we have changed the suggested section (2.4) in the result part to make it easier to understand (line 189-196). “Cytosolic release of sunitinib should result in a change in fluorescence from granules (endocytic vesicles) to diffuse fluorescence from the cytoplasm. The PCI "light after" protocol, with the photochemical treatment applied after sunitinib incubation, showed little difference in the fluorescence pattern of sunitinib upon light exposure, indicating poor cytosolic release with this procedure. This was in contrast to the PCI "light first" protocol where a pronounced cytosolic fluorescence from sunitinib was observed upon light exposure. Thus, PCI with the "light first" procedure was indicated to release sunitinib from endosomal compartments, while PCI with the "light after" procedure was not.”

Paragraph 4.6 has also been made more readable (line 564-572). “For evaluation of cytosolic release of sunitinib after PCI "light first" and "light after" procedure, 1.5 ´ 104 HT-29 cells/well were seeded, allowed to attach and treated as described in section 4.4 to mimic the PCI protocols. For the “light after” procedure, cells were incubated with 2 µM sunitinib for 48 hours, and co-incubated with 0.4 µg/ml TPCS2a for 18 hours. The cells were washed twice and chased 4 hours in drug-free medium before light exposure. In the “light first” procedure, the cells were incubated with 0.4 µg/ml TPCS2a for 18 hours, washed twice and chased 4 hours before light exposure. Sunitinib, 6 µM, was added immediately after light exposure. Image acquisition after light exposure was performed at least an hour after illumination to allow cytosolic release of the endo/lysosomal content.”

13.Paragraph 2.5: how Sun resistance in HT-29/SR correlates with proliferative capacity? (Fig. 3 c). Please add some comments in the result section.

The following has now been added in paragraph 2.5 (line 224-228). "The proliferation rate of HT-29/SR and HT-29 in the presence of sunitinib was also investigated (Figure 3c). A 2 µM sunitinib incubation for 120 hours resulted in 42 % confluence in the HT-29/SR cells while only 31% confluence was observed in the parental HT-29 cell line. The same tendency was shown for all sunitinib concentrations tested. Thus, the proliferation rate of HT-29/SR cells in the presence of 1-10 µM sunitinib was increased compared to the parental cells..”

14.3e and caption: please explain better the significance of “continuous”. Indicate the exact incubation time. In general, the paragraph explaining Sun accumulation results is not well presented.

We have now modified the text in paragraph 4.8 (line 606-607). The aim of the experiment was to evaluate the cellular accumulation of sunitinib in HT-29 parental cells as compared to HT-29/SR cells that had been exposed to sunitinib for up to 5 months. The HT-29/SR cells where kept on sunitinib throughout the project and in 3c they were also seeded in the presence of 2 µM sunitinib and kept on sunitinib for an additional 72 hours.

15.Paragraph 2.5: line 213-215. Please show the data together with that shown in Fig. S4.

We thank the reviewer for pointing out an error here. We have revised the manuscript where “data not shown” is removed. The data is displayed in Supplementary Figure S5.

16.Paragraph 2.7: basing on the completely different condition/environment/drug distribution between cells cultured in monolayers and xenografted in athymic mice, while HT29/SR cells have been excluded a-priori (rationale: no PCI effect on Sun resistance observed in vitro) for in vivo studies? Please comment on this.

The present report indicate another mechanism than lysosomal sequestering responsible for resistance in the HT-29/SR cells, and we therefore concluded that the rational to continue with sunitinib-PCI in HT-29/SR xenografts was not present. We are unsure how the factors listed by the reviewer should interfere with the acquired cellular mechanisms of resistance in HT-29/SR cells.

17.Paragraph 2.8, Fig 5 d. In the CT26.WT mice model the effect on tumor size of PCI+light is quite important, especially at day 4 and basically higher than that of Sun. TPCS2a acts as a pure photosensitizer rather than only as a drug involved in lysosome destabilization.

The reviewer is absolutely right that PCI-sunitinib is a combination treatment between photosensitizer+light and sunitinib. The observed growth delay with TPCS2a and light only shows that we are in a dose range where we expect to find synergy with sunitinib if we have efficient drug release. The light dose used here for CT26.WT tumors, 10 J/cm2, was selected based on previous studies in our group (Weyergang et al., J Control Release 2018, ref 35). This light dose was optimized in BALB/c animals bearing CT26.CL25 tumors in order to detect any effect of combination treatment. Although the photochemical treatment (PS + light) alone had some effect on survival, the combination treatment (PCI) resulted in considerable longer survival, as opposed to the results shown in this present study.

18.Differently, in the in vitro studies using red light a maximum of 20-30% viability reduction was related to PS+ light. Please comment on this.

In the in vitro studies presented for the HT-29 cells the viability after PS+light is reduced to 20 - 50 % depending on the exact experiment. In the in vivo experiments, the TPCS2a+light-treated HT-29 xenografts at day 6 are 60 % of the non-treated controls (Figure 4f), while TPCS2a+light treated CT26 xenografts are 50 % of the untreated controls at day 4 (Figure 5d). The observed tumor growth delay is probably reflecting mechanisms in the tumor stroma as well as in the tumor parenchyma cells. Nevertheless, as we recognize the in vitro and in vivo efficacy of TPCS2a+light to be within a comparable range with modest efficacy.

19.Paragraph 4.4 and 4.5 must be revised to better explain “light after” and “light first” protocols, and probably jointed.

Paragraph 4.4 and 4.5 has been carefully revised and modified as suggested.

Paragraph 4.6: why HT29/SR have been seeded in the presence of Sun?

We have included a sentence explaining this in paragraph 4.6 (line 562). We wanted to evaluated the intracellular localization of sunitinib in HT-29/SR cells. The cells were seeded in the presence of sunitinib in order to maintain sunitinib accumulation. In Supplementary Figure S5 we show that a 24 hours drug-free period in HT-29/SR cells results in the same amount of sunitinib accumulation as the parental cells, despite being continuously cultivated and incubated with sunitinib.

It is based on this observation in Supplementary Figure S5, together with the rest of the results in our manuscript, that has led us to the conclusion that other mechanisms than solely lysosomal sequestration are orchestrating the sunitinib resistance in HT-29/SR.

21.It is unlikely that the resistance can be lost during seeding time.                 

Please see answer to question 20.

22.Control all superscript characters in the text.

We have now carefully gone through the manuscript and made corrections as requested

Reviewer 2 Report

This study by Wong et al. investigates if PCI of sunitinib can enhance the effectiveness of sunitinib entrapped into endo/lysosomal compartments of HT-29 and CT26.WT cell lines. This work also tries to investigate the mechanisms underlying the resistance to therapy. The in vitro cytotoxic outcome of the most effective sunitinib-PCI protocol was also compared to the one achieved using in vivo models. Although molecular mechanisms leading to resistance to sunitinib still remain to be elucidated, in general, experimental planning and sequence, reported data and the analyses appear thorough and solid, the structure and language of the manuscript are adequate. The manuscript fits the scope of Cancers, has merit, is of interest for the field and might be considered for publication, as soon as, from my point of view, the following issues have been addressed:

Major:

Could the release of sunitinib after light illumination be quantified? With the magnification of the fluorescence images included is not possible to determine the % of drug release after light illumination. An accurate methodology could be to isolate lysosomes and quantify the amount of drug inside using both the light before and after protocols and compare the data with non-illuminated samples. In relation to sunitinib quenching effect, it does not always mean that the activity of the drug is compromised. Maybe the fluorescence properties of sunitinib are decreased after illumination but not its activity. Could authors perform a test to measure the activity of sunitinib before and after illumination to validate their conclusion? Moreover, these experiments were performed using blue light illumination, but the in vivo studies were performed using a 652nm laser. Does red illumination also quench sunitinib? Please include these results. Some of the co-localization studies were not performed using confocal microscopy (i.e. Fig 2). This limitation should be addressed in the manuscript. The % of overlapping between the photosensitizer, sunitinib, and Lysotracker should be quantified using, for example, Image J software (line 74 is vague: “The fluorescence was to a large degree overlapping….). Quantification should also be included for Fig 2e. Please clarify why for Fig 3e parental cells were incubated for 72h with sunitinib and HT-29/SR cells were continuously incubated with the drug. In order to compare flow cytometry uptake data, both cell lines should be incubated for the same period of time with the drug. Regarding in vivo studies, there is a minor effect on overall treatment response/survival (Kaplan-Meier plots), but there is a significant reduction of tumor size. What can drive this discrepancy? What is the cause of difference between these two parameters? If authors have carried out PCI studies where the administration of sunitinib was performed later than 30min after light exposure, these results should be included in the revised manuscript. If these experiments were not performed, the reason should be included as it is also mentioned in the discussion.

Minor:

The current title does not seem appropriate. It needs to be revised to better represent the results, particularly taking into account lines 379-380 (PCI was therefore unlikely to release and potentiate sunitinib that was accumulated in the in endo/lysosomal compartments in HT-29/SR cells). The lack of proteomic analysis or studies of molecular pathways leading to resistance to sunitinib should at least be addressed and acknowledged as limitation of the study - – e.g. in the discussion section. Fig 1 appears in the manuscript before an explanation is given for results included in Figs 1F and G. The same is the case for section 2.6 (Figs 3f-I). Fig S6 is not mentioned in the manuscript. Include J/cm2 details for in vitro experiments. Fig 2B: Should the labeling be “light before PCI”? Significant differences are missing in graph 5D but they are mentioned in the text (line 313). Please modify the graph accordingly. Check spelling mistakes. For example: line 82 (eure); line 195 was --> were; line 205 sensitivity in HT-29 cells are --> is. Methodology (several cases): 104cells/well --> 104

Author Response

Reviewer 2

We thank the reviewer for a thorough process which clearly has improved our manuscript

Major:

1.Could the release of sunitinib after light illumination be quantified? With the magnification of the fluorescence images included is not possible to determine the % of drug release after light illumination. An accurate methodology could be to isolate lysosomes and quantify the amount of drug inside using both the light before and after protocols and compare the data with non-illuminated samples.

Please see answer to question 4.

In relation to sunitinib quenching effect, it does not always mean that the activity of the drug is compromised. Maybe the fluorescence properties of sunitinib are decreased after illumination but not its activity. Could authors perform a test to measure the activity of sunitinib before and after illumination to validate their conclusion?

Please see answer to question 4.

Moreover, these experiments were performed using blue light illumination, but the in vivo studies were performed using a 652 nm laser. Does red illumination also quench sunitinib? Please include these results.

The absorption maximum for sunitinib is in the blue region of the visible spectrum (λmax = 429 nm), and no absorption is observed in the red region (Supplementary Figure S4 and Nowa-Sliwinska et al., Cell Death Dis 2015, ref 22). As described in paragraph 2.3 our results indicate that ROS generated by the photochemical reaction (initiated with both blue and red light) target sunitinib. Please also refer to our response to question 11 raised by the first reviewer.

Some of the co-localization studies were not performed using confocal microscopy (i.e. Fig 2). This limitation should be addressed in the manuscript. The % of overlapping between the photosensitizer, sunitinib, and Lysotracker should be quantified using, for example, Image J software (line 74 is vague: “The fluorescence was to a large degree overlapping….). Quantification should also be included for Fig 2e.

We do agree with the reviewer that the suggested experiments in question 1, 2 and 4 would be interesting and further elucidate the mechanism initiated by sunitinib-PCI. However, the aim of the present study was to evaluate sunitinib-PCI as a novel therapeutic approach, and we conclude that the overall treatment response is minor and less than expected. As the suggested experiments will not change this conclusion, it is our opinion that the these experiments should rather be included in future projects on PCI with more suited drugs for cytosolic release.

Please clarify why for Fig 3e parental cells were incubated for 72h with sunitinib and HT-29/SR cells were continuously incubated with the drug. In order to compare flow cytometry

Please refer to our response to question 14 and 20 raised by the first reviewer.

6.Regarding in vivo studies, there is a minor effect on overall treatment response/survival (Kaplan-Meier plots), but there is a significant reduction of tumor size. What can drive this discrepancy? What is the cause of difference between these two parameters?

The tumor size calculations shown in Figure 4f and 4d are assessed at early time point after treatment (6 and 10 days for the HT-29 xenografts, and 4 and 7 days for the CT26 allografts). These results together with the Kaplan-Meier plots indicate that sunitinib-PCI does induce anti-tumor efficacy, but the response is not strong enough to have impact on the overall time to reach endpoint (tumors > 900mm3). This is also in agreement with the H&E and Ki-67 stains in Figure 6a and 6d, showing a larger area of necrosis in sunitinib-PCI treatment tumors as compared to the controls, although the size of the tumors were comparable.

If authors have carried out PCI studies where the administration of sunitinib was performed later than 30 min after light exposure, these results should be included in the revised manuscript. If these experiments were not performed, the reason should be included as it is also mentioned in the discussion.

We believe the suggested addition is already included in the discussion part in line 438-440 as well as in line 477-471. PCI with the light first protocol is most efficient within 2-3 hours after light exposure (Prasmickaite et al., British J Cancer 2002, ref. 18). Here, administration of sunitinib was initiated 30 min after light exposure. The estimated time for tumor accumulation of sunitinib upon per oral administration is 2-4 hours. Thus, the protocol, as applied here, is designed to reach the therapeutic window of the PCI light first procedure. Administration of sunitinib at later time points after light exposure will probably be too late to benefit from PCI.  

Minor:

The current title does not seem appropriate. It needs to be revised to better represent the results, particularly taking into account lines 379-380 (PCI was therefore unlikely to release and potentiate sunitinib that was accumulated in the in endo/lysosomal compartments in HT-29/SR cells).

Here, we do not fully agree with the reviewer that the title should reflect on sunitinib resistance. However, we have now modified the title to better reflect the reported results. "Photochemical-induced Release of Lysosomal Sequestered Sunitinib - Obstacles for therapeutic efficacy"

The lack of proteomic analysis or studies of molecular pathways leading to resistance to sunitinib should at least be addressed and acknowledged as limitation of the study - – e.g. in the discussion section.

In the revised manuscript the following has been included in the discussion part (line 424-425): "Further analysis on molecular pathways controlling cellular sunitinib-resistance is needed to conclude on the mechanisms of sunitinib-resistance in the HT-29/SR cells".

Fig 1 appears in the manuscript before an explanation is given for results included in Figs 1F and G. The same is the case for section 2.6 (Figs 3f-I).

The reviewer is right that these figures appear before they are described in the text. On the other hand there is consistency between the numbering of figures and their explanation in the result part. We have moved both Figure 1 and 3 in the revised manuscript.  

Fig S6 is not mentioned in the manuscript.

We thank the reviewer for drawing our attention to this. A reference to Supplementary Figure S6 (now Supplementary Figure S7) is now included in line 333-334. "The normalized growth curves indicated a response at early time points (Supplementary Figure S7)”.

Include J/cm2 details for in vitro experiments. Fig 2B: Should the labeling be “light before PCI”?

The reviewer is absolutely right and we thank the reviewer for pointing out this error in Figure 2b. The labeling has been changed to “Sun-PCI light first” in the revised manuscript. Furthermore, we have modified the x-axis on Figure 3f from “light dose (seconds)”, which is an incorrect designation, to “light exposure (seconds)”. Regarding light dose, we would like to keep this in seconds instead of J/cm2 as the light sources used in vitro are broad spectrum and not monochromatic light. Instead, we have indicated X second ≈ Y J/cm2 in the figure legends.

Significant differences are missing in graph 5D but they are mentioned in the text (line 313). Please modify the graph accordingly.

The significant differences in graph 5d is presented in Figure 5e.

Check spelling mistakes. For example: line 82 (eure); line 195 was --> were; line 205 sensitivity in HT-29 cells are --> is. Methodology (several cases): 104cells/well --> 104

We have carefully gone through the manuscript and corrected spelling mistakes, including those mentioned by the reviewer.

Reviewer 3 Report

The manuscript from Wong and coworkers describes a through and interesting study of a possible use of photodynamic therapy (PDT) to bolster the activity of a tyrosine kinase inhibitor, sunitinib, in sunitinib resistant cancer cells. The idea behind this strategy is new and is based on the observation that both the photosensitizer, meso-tetraphenyl chlorin disulfonate (TPCS2a), localize in the endosomes and lysosomes of cancer cells. Because the localization of sunitinib in these organelles is thought to play a role in sunitinib resistance, photodynamic activation of TPCS2a concurrently localized in the same subcellular structure, could disrupt the membranes of the organelle, leading to a release of trapped sunitinb, which could then act within the cell in its expected function as a tyrosine kinase inhibitor. The authors present a series of carefully designed experiments that show only an additive effect to combining sunitinib with a TPCS2a-based photodynamic therapy when the photodynamic therapy (PDT) is after therapy with sunitinib. This effect is not a result of a release of sunitinib from lysosomes but rather a result of its destruction within the lysosome; the authors argue that for the “light after” use of PDT, that reactive oxygen species within the lysosomes produced during PDT are chemically degrading the sunitinib also trapped within the same organelle. They provide evidence (UV-vis) that the combination of PDT and sunitinib in PBS + 1% FCS solution leads to a destruction of sunitinib. For the “light before” scheme, a synergistic effects is apparent and less sunitinib is indeed trapped in the lysosomes. However, this in vitro synergistic effect could not be confirmed in two mice tumor models.

Although the results disprove the original working hypothesis, this is a well constructed study that illustrates the pitfalls of rationally designed combination cancer therapies. The work will be of interest to pharmacologist and oncologist working in the area of tyrosine kinase inhibitors and PDT. I recommend publication after the following points have been addressed.

The name of the photosensitizer should be reported in the abstract.

It would be helpful to see the structures of sunitinib and TPCS2a in the introduction.

The stability studies of the combinations of sunitinib and PDT were done in solutions of PBS (pH = 7.4) containing 1% FCS. However, in the cells these reactions would be taking place in the acidic medium of the lysosome. Thus, the author should repeat these studies at a lower pH, i.e. 4-5, to be sure the stability is not increased under acidic conditions. If this were the case, the interpretation of the study needs to be reevaluated.

Drug concentrations, light-doses and treatment times should be reported in all figures (e.g. 1G, S3 and S4) so one doesn’t have to keep looking through the experimental section to find this information.

Lines 215-217. Why wash out sunitinib before treatment with sunitinib? Washing out should take place after treatment.

Author Response

Reviewer 3 

We thank the reviewer for a thorough process which clearly has improved our manuscript

The name of the photosensitizer should be reported in the abstract. The name of the photosensitizer (is now included in the abstract

The name of the photosensitizer disulfonated tetraphenyl chlorin (TPCS2a) is now included in the abstract line 22-23 in the revised manuscript: "By super-resolution fluorescence microscopy, sunitinib was found to accumulate in the membrane of endo/lysosomal compartments together with the photosensitizer disulfonated tetraphenyl chlorin (TPCS2a). Furthermore, the treatment effect of sunitinib was potentiated by PCI...”.

It would be helpful to see the structures of sunitinib and TPCS2a in the introduction.

We thank the reviewer for this suggestion and have now included the structures of both sunitinib and TPCS2a in the supplementary file (Supplementary Figure 1).  

The stability studies of the combinations of sunitinib and PDT were done in solutions of PBS (pH = 7.4) containing 1% FCS. However, in the cells these reactions would be taking place in the acidic medium of the lysosome. Thus, the author should repeat these studies at a lower pH, i.e. 4-5, to be sure the stability is not increased under acidic conditions. If this were the case, the interpretation of the study needs to be reevaluated.

We thank the reviewer for this suggestion. Absorption spectra for sunitinib in the presence of TPCS2a before and after light exposure has now been assessed also at pH 5. The results show that the photochemical treatment attenuates the light absorption of sunitinib also at pH 5. However, we do not agree that the experiments on pH = 5 necessarily is the best model for studying photochemical damage of sunitinib in endocytic vesicles. Photochemical targeting of the endosomal membrane has previously been shown to release H+ from the endosomal lumen and thereby increase the endosomal pH prior to cytosolic release of endosomal content (Ohtsuki et al., Sci Rep 2015, ref 27). The absorption spectra for pH 5 has now been included in Supplementary Figure S4, and are discussed in line 142-146 as well as in line 400-406. Furthermore, the material and method section related to spectroscopy has also been modified.

Drug concentrations, light-doses and treatment times should be reported in all figures (e.g. 1G, S3 and S4) so one doesn’t have to keep looking through the experimental section to find this information.

In the revised version of our manuscript this has been corrected as suggested.  

Lines 215-217. Why wash out sunitinib before treatment with sunitinib? Washing out should take place after treatment

Please find the response to question 20 raised by the first reviewer .